# Local cation-tuned reversible single-molecule switch in electric double layer

Ling Tong[1,3], Zhou Yu[1,3], Yi-Jing Gao[1,2,3], Xiao-Chong Li[1], Ju-Fang Zheng[1], Yong Shao [1], Ya-Hao Wang [1] ✉ & Xiao-Shun Zhou [1] ✉

The nature of molecule-electrode interface is critical for the integration of atomically precise molecules as functional components into circuits. Herein, we demonstrate that the electric field localized metal cations in outer Helmholtz plane can modulate interfacial Au-carboxyl contacts, realizing a reversible single-molecule switch. STM break junction and I-V measurements show the electrochemical gating of aliphatic and aromatic carboxylic acids have a conductance ON/OFF behavior in electrolyte solution containing metal cations (i.e., $Na^+$, $K^+$, $Mg^{2+}$ and $Ca^{2+}$), compared to almost no change in conductance without metal cations. In situ Raman spectra reveal strong molecular carboxyl-metal cation coordination at the negatively charged electrode surface, hindering the formation of molecular junctions for electron tunnelling. This work validates the critical role of localized cations in the electric double layer to regulate electron transport at the single-molecule level.

With the increasing demand for the miniaturization of electronic devices, the use of chemically identical nanometer-sized molecules as functional components has become a thriving subfield of nanoscience[1,2]. The construction of a reliable molecular switch is an important step because of its key role in information storage, logical data manipulation, and signal processing[3,4]. Over past three decades, a common approach to design single-molecule switch is to use a molecular backbone that can change the molecular conformations or spin/redox states upon exposing to external stimulus, such as light[5,6], mechanics[3,7,8], pH[9,10], chemical reactants[11], magnetism[12–15], and electricity[16–22]. However, this often requires complex organic synthesis to obtain a stimuli-responsive molecular scaffold that can switch the tunnel junctions between two stable conductance states. Most of the external stimulus methods cannot be used in applications that require all-electric drive circuit components[23]. Alternatively, interfacial molecule–metal contacts can also significantly determine electron transport through the molecular junctions[24–27]. To date, the pre-setting anchoring groups in molecular backbones has been the typical way to

manipulate the molecule–electrode interaction and electron transport. However, in situ control of the molecule–electrode contacts toward electrical functional components in the same molecular backbone remains technically challenging.

Electrochemical gating is one of the most feasible and effective methods to tune interfacial electronic structures, which has been widely applied in studying electron transport across single molecules as a function of electrode potential in different electrolyte solutions[18,28,29]. So far, the research reports with electrochemical gating have mainly focused on regulating electron transport by altering the energy alignment between the Fermi level of electrodes and molecules[18,28,29], ignoring the potential-dependent structure of electric double layer might change the molecule–electrode contacts. This arises from the most used molecular anchoring groups, such as thiols, can be covalently bonded to electrodes with little effect on the electrochemical potentials. The carboxylic acid group is one of the most used anchoring groups to fabricate molecular junctions[30–32]. Recently, break-junction measurements have proven that the formation of

[1]Key Laboratory of the Ministry of Education for Advanced Catalysis Materials, Institute of Physical Chemistry, Zhejiang Normal University, 321004 Jinhua, China. [2]Zhejiang Engineering Laboratory for Green Syntheses and Applications of Fluorine-Containing Specialty Chemicals, Institute of Advanced Fluorine-Containing Materials, Zhejiang Normal University, 321004 Jinhua, China. [3]These authors contributed equally: Ling Tong, Zhou Yu, Yi-Jing Gao. ✉e-mail: yahaowang@zjnu.edu.cn; xszhou@zjnu.edu.cn

carboxyl-linked molecular junctions relies on the interaction through deprotonated −COO− groups[30,31,33]. In addition, it is reported that the deprotonated −COO− groups can coordinate with the alkali metal ions and transition metal ions[34,35]. Meanwhile, electrochemical potentials can effectively change the charge states of the electrode surface to tune the distribution of ions in the electric double layer at the interfaces, which in turn affect the interfacial dissociation equilibrium of −COOH ⇌ H+ + −COO− and carboxyl-metal cation interactions. This may provide a unique opportunity to realize reversibly controlled molecule−metal contacts for tuning electron transport, but has not been explored and reported so far.

To test the above-mentioned hypothesis, herein, we have employed in situ STM break junction (STM-BJ) and Raman spectroscopy to probe the electrochemical potential-dependent molecular conductance and structures of carboxylic acids molecules with methyl sulfide (SMe) adsorbed on Au(111) electrode. In the electrolyte solution containing metal cations (i.e., Na+, K+, Mg2+, and Ca2+), single-molecule electrical measurements shows the conductive states of aliphatic and aromatic carboxylic acids molecular junctions transform from a detectable state (ON state) at 0 V to below the minimum detection limit of the currently used amplifier range (OFF state) at −0.5 V vs. Pt. Cyclically switching the applied potentials in both STM-BJ and I−V tests leads to the conductance ON/OFF ratio of 4-(methylthio) benzoic acid (4-MTBA), 3-(methylsulfanyl) propanoic acid (MPA) and terephthalic acid (TPA), compared to almost no conductance change without the metal cations in solutions. In conjunction with Raman spectra and DFT simulations, it is found that the adsorbed molecules change from carboxyl-metal cation coordination, protonation to deprotonation, as the applied potential changes the electrode surface from negatively to positively charged. Therefore, a reversible single-molecule switch could be realized in the carboxylic acid-linked junctions through electrochemical control of the localized ions.

## Results

### Effects of electric field-localized cations on single-molecule junctions

The main idea of our work is to exploit the electric field-localized cations in outer Helmholtz plane (OHP) to control the forms of carboxylic acid molecules (i.e., protonated or deprotonated, carboxyl-metal cation coordination) assembled on Au(111) substrate, which can significantly affect the Au−carboxyl contacts with the Au STM tip during the formation of molecular junctions. By controlling the electrochemical potential with a potentiostat, the electrode surface can be easily charged positively or negatively. Then the anions or cations in the solution can be concentrated in OHP as counter ions, compensating the negatively or positively charged electrode surface according to Gouy−Chapman−Stern model. As schematically illustrated in Fig. 1a, the local hydrated protons can protonate the self-assembled monolayers (SAMs) of carboxylic acids molecules with methyl sulfide (SMe) adsorbed on Au(111), and hydrated metal cations (i.e., Na+, K+, Mg2+, Ca2+) can coordinate with the −COO− groups at negatively charged Au(111) electrode. These might hinder the formation of Au−COO− contacts when an Au STM tip is driven into and suspended above SAMs. In contrast, at the positively charged Au(111) surface, the local hydroxide anions can deprotonate the SAMs, and the electrostatic repulsion can disrupt the carboxyl-metal cation coordination, leaving abundant of −COO− groups that could bind to Au tip to form molecular junctions for electron transport.

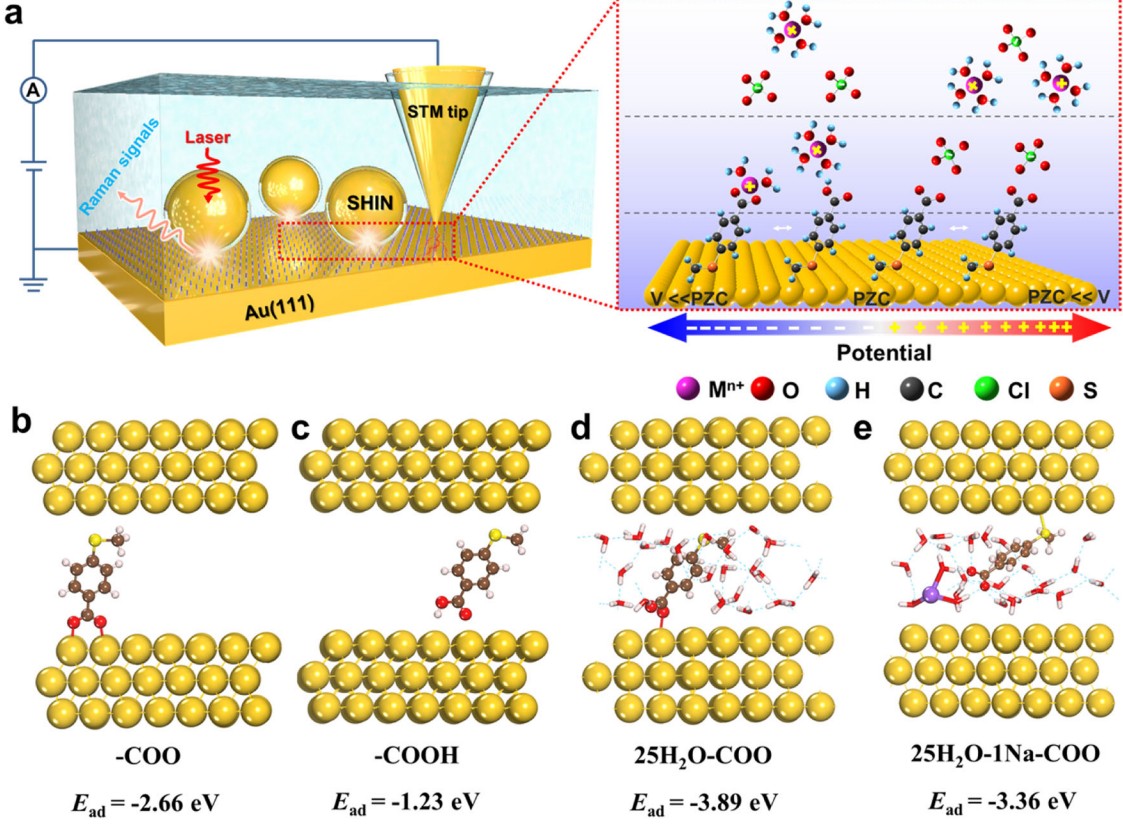

**Fig. 1 | Electric field-localized cations to control Au−carboxyl contacts.**
**a** Schematic of the interfacial model of carboxylic acids molecules with SMe adsorbed on Au(111) surface at different gate potentials probed by STM-BJ and Raman spectroscopy. PZC, the potential of zero charge. The optimized structures of (**b**) deprotonated 4-MTBA with COO− on Au(111), **c** protonated 4-MTBA with COOH on Au(111), **d** deprotonated 4-MTBA with COO− on 25H2O/Au(111) and **e** deprotonated 4-MTBA with COO− on Na-25H2O/Au(111).

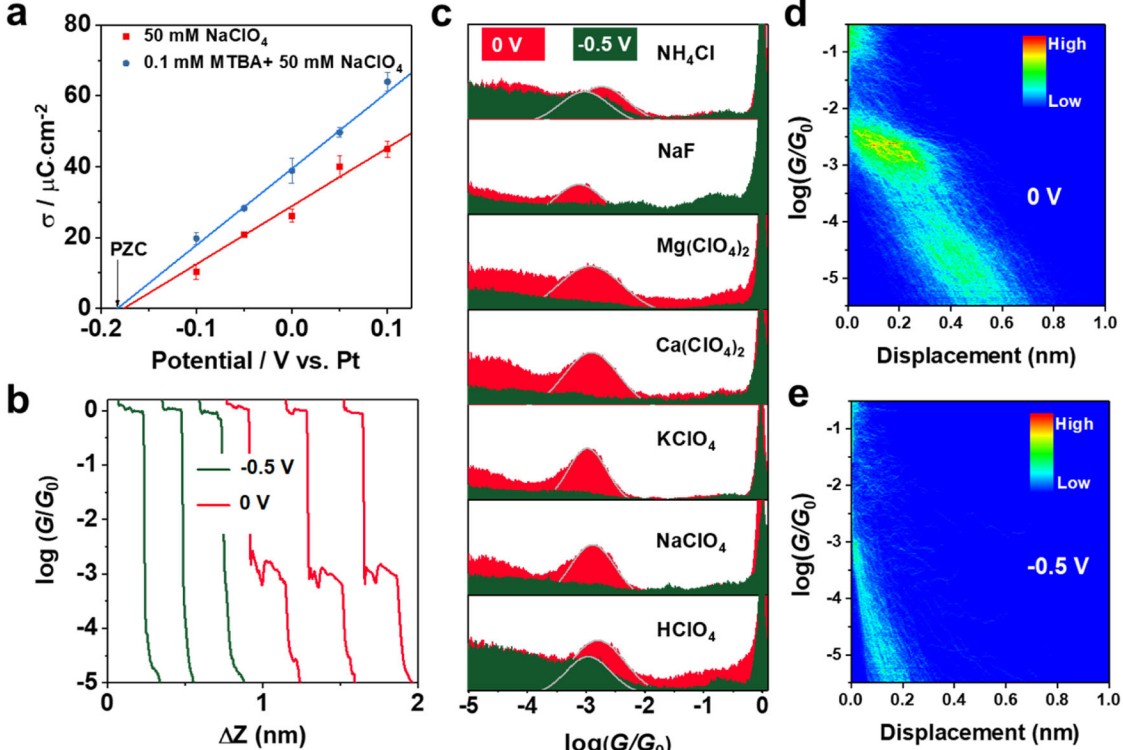

**Fig. 2 | Single-molecule conductance measurements on positively or negatively charged Au(111) surface. a** The charge densities derived from integration of the immersion current transients against the applied potentials of Au(111) in 50 mM NaClO₄ without or with 0.1 mM 4-MTBA. Error bars represent standard deviation obtained from different experiments. **b** Typical conductance traces of single-molecule conductance measurements obtained on Au(111) substrate in 0.1 mM 4- MTBA + 50 mM NaClO₄ solution with potential control at 0 V or −0.5 V. **c** 1D conductance histograms of 4-MTBA obtained in different electrolyte solutions, and 2D conductance-displacement histograms of 4-MTBA in 50 mM NaClO₄ solution obtained at **d** 0 V and **e** −0.5 V, respectively. The counts are normalized by the number of conductance traces used. All potentials are specified with respect to Pt.

To substantiate this hypothesis, we first performed density functional theory (DFT) and ab initio molecular dynamics (AIMD) calculations to analyze the interaction between carboxylic acid molecules and Au electrode in different configurations using the Vienna Ab Initio Simulation Package (VASP) software (see "Methods" and Supplementary Figs. 1 and 2 in Supplementary Information). Figure 1b–e shows the optimized structures of different configurations of 4-MTBA adsorbed on a three-layer (2×2) Au(111) surface. The adsorption of protonated 4-MTBA with COOH ($E_{ads}$ = −1.23 eV) is significantly weaker than that of deprotonated 4-MTBA with COO⁻ ($E_{ads}$ = −2.66 eV). Similarly, the deprotonated 4-MTBA adsorb on the 25H₂O/Au(111) through −COO⁻ group, whose two O atoms strongly bind to surface Au. When adding Na⁺, the hydrated metal cation makes −COO⁻ more than 4 Å from the Au surface on the Na-25H₂O/Au(111), resulting in weaker adsorption. These support the feasibility of controlling Au−carboxyl contact by local cations.

Next, we measured the potential of zero charge (PZC) by immersing a dry and clean Clavilier-type Au(111) half-bead single-crystal electrode in electrolyte solution with potential control following the reported papers[36]. The immersion $I$–$t$ curves recorded using a potentiostat can be found in Supplementary Fig. 3. The corresponding charge density with respect to the applied potentials is shown in Fig. 2a, the intersection of the extrapolation with the X-axis is PZC, ca. −0.18 V vs. Pt for bare Au(111) in 50 mM NaClO₄. This corresponds to 0.3 V vs. SCE, very close to the reported value of 0.285 V for Au(111) electrode in the same solution[37]. As adding 0.1 mM 4-MTBA in the solution, the PZC negatively shifts by about 10 mV because of the molecular adsorption. When changing the electrolyte to HClO₄ or Ca(ClO₄)₂, a similar PZC could be found (Supplementary Fig. 4).

Then, single-molecule conductance measurements of 4-MTBA were carried out by using electrochemical STM-BJ with potential controls. Experimental details could be found in "Methods". The potential of Au(111) electrodes was firstly set at 0 V vs. Pt in 0.1 mM 4-MTBA + 50 mM NaClO₄, which is higher than that of PZC, indicating a positively charged surface. Figure 2b shows the obtained representative conductance-displacement traces. The step feature appeared at about $10^0$ $G_0$ ($G_0 = 2e^2/h$, where $e$ is the electron charge and $h$ is Planck's constant) is assigned to the formation of Au atomic contacts during the stretching process of the tip[38,39]. As the tip further stretches, there is another step feature appeared at $10^{-3.0}$ $G_0$ after rupturing Au atomic contacts (red curves). This corresponds to the formation of molecular junctions of 4-MTBA at the positively charged Au surface. Thousands of these conductance-displacement traces are collected to construct 1D conductance histograms, showing an obvious conductance peak at around at $10^{-3.0}$ $G_0$ in the one-dimensional (1D) conductance histograms in Fig. 2c (red peak). The statistical analysis of displacements in all conductance-displacement curves from $10^{-6.0}$ to $10^{-0.3}$ $G_0$ show an obvious stretching state centered around $10^{-3.0}$ $G_0$ after rupturing Au atomic contacts in the two-dimensional (2D) conductance histogram (Fig. 2d), the step stretching displacement ($\Delta z$) distribution with a Gaussian-fitted peak at 0.4 nm (Supplementary Fig. 5). By adding the snapback distance (0.5 nm) of breaking Au−Au contacts[27,28], the most probable absolute displacement for molecular junctions is about 0.9 nm, which is in good agreement with the molecular length of 4-MTBA. These further confirm the formation of single-molecule junctions.

When an electrochemical potential of −0.5 V vs. Pt was applied on the Au(111) substrate to negatively charge the surface, interestingly,

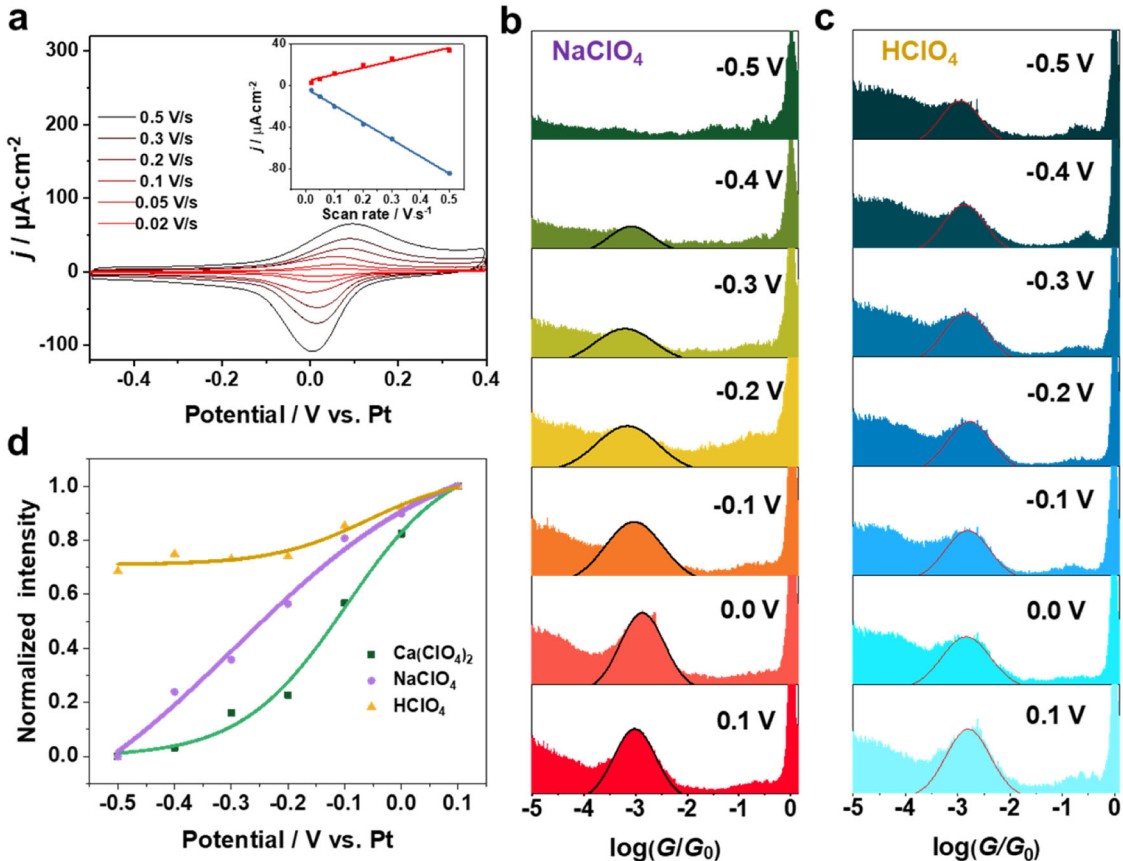

**Fig. 3 | Electrochemical gating of single-molecule conductance in the presence of different cations. a** CVs of Au(111) obtained in 0.1 mM 4-MTBA + 50 mM NaClO$_4$ solution with different scan rates, insert is the plot of the current density of the anodic and cathodic peaks against the scan rates. The potential-dependent 1D conductance histogram of 4-MTBA in 50 mM **b** NaClO$_4$ and **c** HClO$_4$. **d** The normalized intensity variations of conductance peaks against the applied potentials in solutions containing H$^+$, Na$^+$, and Ca$^{2+}$ cations, respectively.

these step features and conductance peaks appeared at $10^{-3.0}$ $G_0$ in Fig. 2b, c (green curves) both disappeared. Similar phenomena were also found in other electrolyte solutions containing metal cations of K$^+$, Mg$^{2+}$, and Ca$^{2+}$; While in the solution with non-metal cations (i.e., H$^+$ and NH$_4^+$), these conductance peaks can still be observed in Fig. 2c. The single-molecule conductance of 4-MTBA in a different electrolyte solution and the pH of solution are summarized in Supplementary Table 1. The weak conductance peak at 0 V in NH$_4$Cl solution might arise from the strong specific adsorption of halide ions and hydrolysis reaction (Supplementary Fig. 6). In addition, the cation-tuned conductance switch remains effective when changing the polarity of the bias voltage (Supplementary Fig. 7). Single-molecule conductance measurements of 1,4-bis(methylsulfanylmethyl) with two -SMe groups show obvious conductance peaks at these two potentials (Supplementary Fig. 8). These reveals that the local metal cations in the OHP at negatively charged electrode surface can significantly inhibit Au–carboxyl contacts to form molecular junctions for electron transport.

## Electrochemical gating with different cations

To further disclose these interesting potential-dependent conductance behaviors, we employed electrochemical STM-BJ to measure the single-molecule conductance of 4-MTBA with changing the potentials of Au(111) substrate. A Clavilier-type Au(111) half-bead single-crystal electrode immersed in 0.1 mM 4-MTBA + 50 mM NaClO$_4$ aqueous solution is firstly examined by using cyclic voltammograms (CVs). As shown in Fig. 3a, there is one pair of well-defined reversible peaks at about 0.05 V. Due to the good stability of carboxylic acids, no faradaic

processes occur in this electrochemical potential range. Previous reports assigned them to deprotonation or protonation of the carboxyl groups in positive or negative scans of the electrochemical potential[40,41]. A linear correlation between anodic (red square) and cathodic (blue square) current peaks with the scan rates is found in the insert. This proves that the reversible peaks arise from the interfacial 4-MTBA assembled on the Au(111). In addition, such reversible current peaks can also be observed in the CVs using SCE and Pt as the reference electrodes in 50 mM NaClO$_4$ and 50 mM HClO$_4$ solutions (Supplementary Fig. 9), respectively. The coverage of adsorbed molecules is quantitatively estimated from the total charges of anodic current peaks, 14.3 μC/cm$^2$ is 1.5 × 10$^{-10}$ mol/cm$^2$ for 4-MTBA, comparable to the reported results[41].

In line with CV profile, the conductance measurements were carried out from −0.5 to 0.1 V vs. Pt in 0.1 V increments. As shown in Fig. 3b, a conductance peak near $10^{-3.0}$ $G_0$ appears gradually and becomes intense as the potential increases. Such potential-dependent characteristics of conductance peaks can also be observed in the solutions containing other metal cations of K$^+$, Mg$^{2+}$, Ca$^{2+}$ (Supplementary Fig. 10), and increasing electrolyte concentration make the conductance peak disappear at higher potentials (Supplementary Fig. 11). We also performed the potential-dependent conductance measurements in 50 mM HClO$_4$ without the metal cations. It is found that the conductance peak becomes weak but does not disappear at −0.5 V (Fig. 3c). This arises from the populations of deprotonated molecules become less, resulting in a lower formation probability of molecular junctions[33]. While changing molecules to 1,4-bis(methylsulfanylmethyl), the applied potentials and cations have little effect on

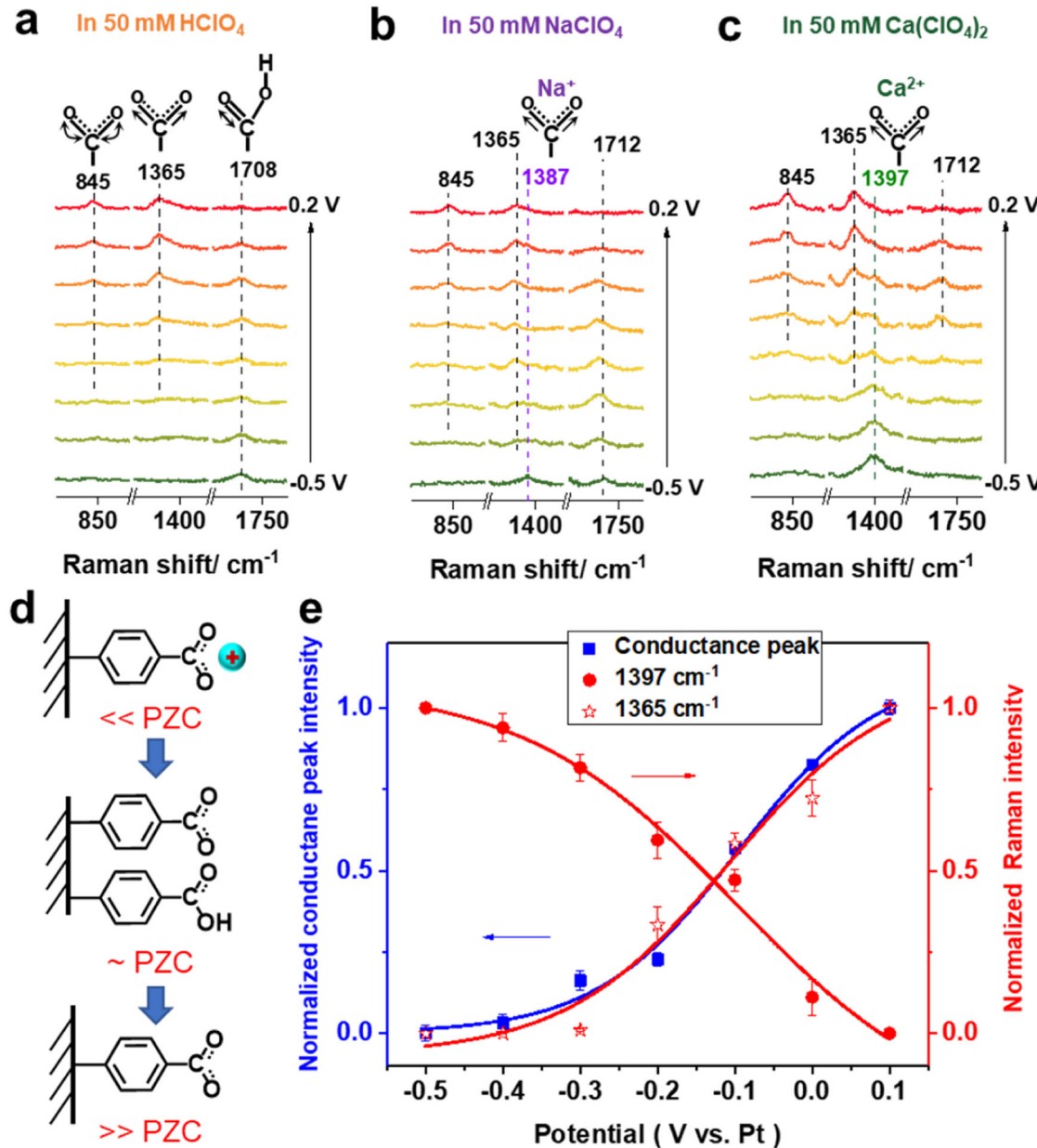

**Fig. 4 | Potential-dependent molecular structures at Au(111)/electrolyte interfaces.** Potential-dependent Raman spectra obtained at Au(111) substrate in 50 mM **a** HClO$_4$, **b** NaClO$_4$, and **c** Ca(ClO$_4$)$_2$ aqueous solution contain 0.1 mM 4-MTBA. **d** Schematic of interfacial 4-MTBA transforms on different charge states of Au(111) surface. **e** The plot of normalized Raman band intensities of $v_s$(COO$^-$Ca$^{2+}$) and $v_s$(COO$^-$), and the conductance peak intensities against the applied potentials of Au (111) substrate. Error bars represent standard deviations obtained from Gaussian fits of conductance peak intensities and Raman intensities.

the conductance peaks (Supplementary Fig. 12). The normalized intensity variations of conductance peaks relative to the applied potentials are summarized in Fig. 3d. This clearly confirms that the concentrated metal cations in OHP at the very negatively charged Au(111) surface, can totally inhibit the formation of Au−carboxyl contacts to block electron transport. Significantly, the conductance peak intensities decrease fast in the presence of the divalent cations of Ca$^{2+}$, which arises from its larger charge acceptability and binding affinity with the carboxyl groups.

**Molecular structures at the electrochemical interfaces**

To verify the mechanism of local cations on Au−carboxyl contacts, shell-isolated nanoparticle-enhanced Raman spectroscopy (SHINERS)[36,42–44] is employed to in situ probe the electrochemical process of 4-MTBA on Au (111). As schematically illustrated in Supplementary Fig. 13, the

120 nm Au core nanoparticles (NPs) coated ca. 2 nm SiO$_2$ shell are dropped on Au(111) to work as the Raman-signal amplifiers. Three-dimensional finite-difference time-domain simulations revealed that the electromagnetic field strength in the gap between NPs and Au (111) surface could be enhanced up to 7−8 orders of magnitude under 633 nm laser illumination for detecting the surface species[36,42]. In addition, the chemically inert silica shells can protect the Au core from the analytes and environments to avoid generating interference Raman signal from the Au core.

Figure 4a shows the potential-dependent Raman spectra acquired at the Au (111) electrode in 0.1 mM 4-MTBA + 50 mM HClO$_4$ with 0.1 V intervals. The 600−1800 cm$^{-1}$ spectral range contains all the most significant peaks of 4-MTBA (Supplementary Fig. 14). At −0.5 V, an obvious band at 1708 cm$^{-1}$ are assigned to C = O stretching of −COOH (v(COOH)), consistent with previous reports[45,46].

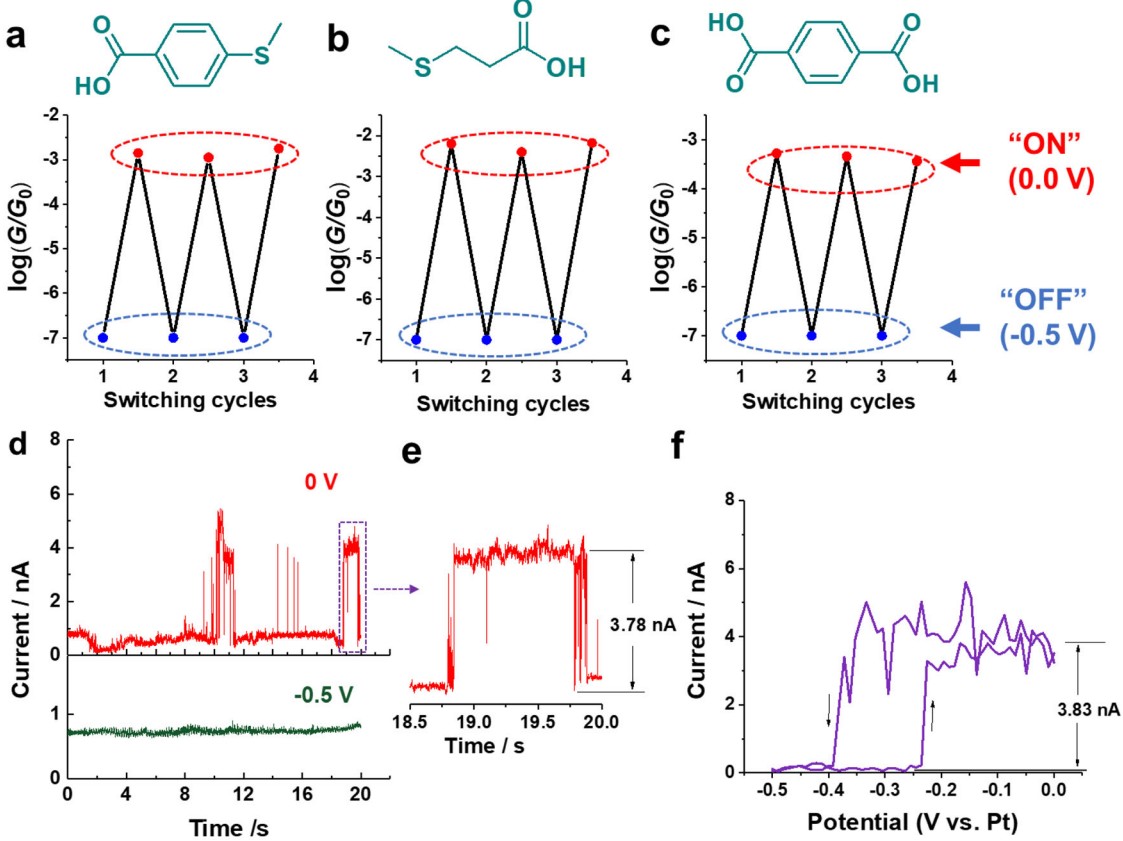

**Fig. 5 | Local-cation-tuned reversible single-molecule switch performance.**
**a** Sequential cycles of potential-induced conductance modulation of **a** 4-MTBA,
**b** MPA, and **c** TPA between "ON" and "OFF" states. The molecular structures are
shown in the top panel. **d** Typical $I-t$ traces in 0.1 mM 4-MTBA + 50 mM NaClO$_4$
solution without feedback loop at the substrate potentials of 0 and −0.5 V,
respectively. Initial tunneling current setpoint is 0.8 nA under a bias voltage of
50 mV. **e** Zoom in $I-t$ trace showing the formation of single-molecule junction
formation. **f** A typical $I-V$ curve obtained in 0.1 mM 4-MTBA + 50 mM NaClO$_4$ with a
constant bias of 50 mV by simultaneously sweeping the tip and substrate potentials
between 0 and −0.5 V.

As potentials increase, it becomes weak, while two bands at 845 and
1365 cm$^{-1}$ ascribed to COO$^-$ bending ($\beta$(COO$^-$)) and symmetric
stretching ($v_s$(COO$^-$)), which appear and become intense above
−0.2 V. These indicate the molecular -COOH groups dissociates into
−COO$^-$. When changing the electrolytes to NaClO$_4$, interestingly, an
obvious Raman band at 1387 cm$^{-1}$ can be observed at −0.5 V in Fig. 4b.
Previous $^{23}$Na NMR experiments show that Na$^+$ ions presented near
the surface play an important role in stabilizing the high coverage of
the carboxylate on the Au surface[34]. In addition, the calculated
Raman spectra of 4-MTBA-hydrated Na$^+$ coordination complex also
present an obvious band at 1396 cm$^{-1}$ related to the C = O symmetric
stretching ($v_s$(COO$^-$Na$^+$)) in Supplementary Fig. 15. Therefore, the
Raman band at 1387 cm$^{-1}$ is ascribed to the $v_s$(COO$^-$Na$^+$), consistent
with reported carboxyl-metal cation complex systems[45]. This proves
that the local Na$^+$ cations can strongly coordinate with the -COO$^-$
groups at negatively charged Au(111) surface. Similarly, a much
intense Raman band ascribed to carboxyl-Ca$^{2+}$ symmetric stretching
($v_s$(COO$^-$Ca$^{2+}$)) blue shift to 1397 cm$^{-1}$, and the Raman band of
$v$(COOH) disappears at −0.5 V in Fig. 4c, which arise from the strong
binding affinity of Ca$^{2+}$ with −COO$^-$ groups. The same variation
characteristics of spectral peaks can also be confirmed in the calcu-
lated spectra in Supplementary Fig. 15.

As the potential of the electrode increases, the Raman intensities
of $v_s$(COO$^-$Na$^+$) and $v_s$(COO$^-$Ca$^{2+}$) decrease to disappear. Meanwhile,
the $v_s$(COO$^-$) band rises and becomes intense; While the Raman
intensity of $v$(COOH) firstly increases and reach maximum around
PZC, then decreases both in Fig. 4b, c. Therefore, we illustrate
interfacial 4-MTBA transforms on different charge states of Au(111)

surface in Fig. 4d. When the applied potential causes the electrode
surface to change from negative to zero to positive charge, the
interfacial 4-MTBA coordinated with the metal cation is firstly con-
verted into protonated and deprotonated molecules, and then all
into deprotonated molecules. Quantitatively, the normalized
Gaussian-fitted Raman band of $v_s$(COO$^-$Ca$^{2+}$), $v_s$(COO$^-$) and the con-
ductance peak intensities against the potentials of Au(111) substrate
are shown in Fig. 4e. It clearly shows that the formation of molecular
junctions relies on the deprotonated −COO$^-$. Compared to in HClO$_4$
solution, the local metal cations can displace the proton in the car-
boxyl group to form the ligand complex at the very negatively
charged surface, which inhibits the Au−COO$^-$ contacts to form
molecular junctions.

## Single-molecule switch performance

To prove such local-cation-tuned reversible single-molecule switches,
we performed single-molecule conductance measurements of 4-MTBA
and cycled the potentials between −0.5 V and 0 V. As shown in Sup-
plementary Fig. 16a, a well-defined conductance peak at $10^{-3.0}$ $G_0$
repeatedly appears at 0 V, but disappears at −0.5 V in the conductance
histograms. Furthermore, other aliphatic and aromatic carboxylic
acids, 3-(methylsulfanyl)propanoic acid (MPA) and terephthalic acid
(TPA), also shows the same interesting switching phenomena upon
changing the potentials from −0.5 V to 0 V (Supplementary Fig. 16b, c).
The switching cycle tests are summarized in Fig. 5a−c with con-
ductance values taken from the conductance histograms constructed
by thousands of conductance-displacement traces with a Gaussian fit.
In addition, a lower single-range current amplifier down to $10^{-7.0}$ $G_0$

(at the current bias of 50 mV) was also used for the break-junction experiments, there is also no conductance peak observed at −0.5 V (Supplementary Fig. 17). The conductance peak disappears due to the strong molecular carboxyl-metal cation coordination on the negatively charged electrode surface, which is considered as the conductance OFF state. Thus, the conductance ON/OFF states might be effectively achieved through the electrochemical control. In addition, cycling tests were also performed by repeatedly cycling the potential scans followed by single-molecule conductance measurements, the conductance peaks can repeatedly appear at 0 V and disappear at −0.5 V over 50 cycles (Supplementary Fig. 18). This demonstrates the good stability of localized cation-tuned reversible single-molecule switches in the electric double layer.

To verify this local-cation control can be applied beyond break-junction experiments, we further performed $I$–$t$ tests without STM feedback loop according to previous reports[47,48]. The experimental details can be found in "Methods". Figure 5d shows typical untreated raw I-traces at different substrate potentials. At 0 V, current blinking can be observed at around t = 10 s, then the current jumps to ON state for lasting more than 1 s, consistent with the previous report[47]. This similar phenomenon repeats around t = 19 s. The current jump height is about 3.78 nA, corresponding to a conductance of $10^{-3.0}$ $G_0$, which is coherent with single-molecule conductance in the STM-BJ measurements. Instead, only the base current and its fluctuations were observed when the substrate potential was controlled at −0.5 V. In addition, the $I$–$t$ tests with a very low STM feedback loop[49–51] have also shown the characteristic electron tunnel-current spikes ascribed to trapped molecules in the nanogap of two electrodes to form a molecular junction at the positively charged electrode surface, rather than at negatively charged electrode surface (Supplementary Fig. 19). These further confirm that the electric field-localized cations at different charged electrode surface can modulate the molecule–metal contacts, leading to conductance ON/OFF states.

To further test the switching effects, the $I$–$V$ measurements have been carried out in 0.1 mM 4-MTBA + 50 mM NaClO$_4$ solution with a constant bias of 50 mV in line with previous reports[52–54]. The potentials of Au(111) substrate are swept between 0 and −0.5 V. Figure 5f shows the typical $I$–$V$ curve recorded upon the formation of molecular junctions. Obviously, the current jumps to a low value or a high value during a negative or positive potential sweep, respectively. These can be attributed to the breakdown and formation of molecular junctions. The relative current difference is about 3.83 nA, comparable to single-molecule conductance in the break-junction measurements. When the gate potential is lower than −0.41 V, tip current is turned off for all $I$–$V$ parallel tests (Supplementary Fig. 20), which is consistent with the disappearance of conductance peak below −0.4 V in single-molecule break-junction experiments. These further prove the local-cation-controlled single-molecule switch. In addition, it is worth mentioning that this local-cation-tuned single-molecule switch depends on the rate at which the applied potential changes the structure of the electric double layer. This might lead to low switching frequency. On the other hand, when the gate potential is changed, the base current including the electric double-layer charging current and the tip leakage current can affect the switching performance, such as ON/OFF ratio.

In summary, a local-cation-controlled reversible single-molecule switch has been successfully demonstrated in carboxylic acid molecular junctions with electrochemical STM-BJ, $I$–$t$ and $I$–$V$ techniques for the first time. With the help of in situ Raman spectroscopy, the switching mechanism is found as follows: At negatively charged Au(111) surface, local concentrated hydrated metal cations (Na$^+$, K$^+$, Mg$^{2+}$, Ca$^{2+}$) in outer Helmholtz plane (OHP) can coordinate with the -COO$^-$ groups of the self-assembled monolayers (SAMs). This inhibits the formation of Au−COO$^-$ contacts for electron transport when an Au STM tip is driven into or suspended above SAMs (OFF state); At positively charged Au(111) surface, the electrostatic repulsion can disrupt the carboxyl-metal cation coordination, and local hydroxide anions can deprotonate the SAMs, leaving abundant of -COO$^-$ groups that can bind to Au tip to form molecular junction for electron transport (ON state). The present work offers a detailed molecular picture of the effect of localized cations on carboxyl molecules at the electrochemical interfaces, paving a new avenue for realizing reversible single-molecule switches via gate electrodes.

## Methods

### Electrochemical measurements

All electrochemical measurements were carried out at a CHI-660E potentionstat using a homemade three-compartment glass cell with Pt as reference electrode and counter electrode. A Clavilier-type Au(111) half-bead single-crystal electrode was used as the work electrode. The solutions in electrochemical measurements were deaerated and protected by argon. The open circuit potential between the Pt wire and SCE is 0.48 V in 0.1 mM 4-MTBA + 50 mM NaClO$_4$ solution.

### Single-molecule conductance measurement

Conductance measurement by using STM break-junction (STM-BJ) technique was performed on the home-modified Nanoscope IIIa STM (Veeco, US). In a four-electrode electrochemical system, the same Pt ring and Pt wire were used as counter electrode and reference electrode in all electrochemical STM-BJ experiments, respectively. The naturally formed (111) facet at an Au bead substrate and mechanically cut Au STM tip (0.25 mm diameter, 99.999%, Alfa Aesar) were used as work electrodes. To minimize the ionic leakage current in electrochemical environments, Au tip is coated with thermoset polyethylene glue with a little exposed tip apex. Prior to each experiment, the substrate was electrochemical-polishing and flame-annealing. The single-molecule conductance measurements were performed in a solution containing 0.1 mM 4-MTBA + 50 mM electrolytes. The molecules can adsorb on Au(111) through the anchoring groups of methyl sulfide (SMe) to form a self-assembled monolayer.

The procedure of the STM-BJ method is briefly described as follows: Firstly, the STM tip is driven toward the substrate to a preset current value (50 nA) via piezoelectric control. Then an external pulse voltage is applied on z-piezo to bring STM tip into the substrate surface to ensure tip contact with the substrate. Then the tip is pulled away from the substrate at a constant speed of 20 nm/s. During the process, molecular junctions can be formed. Meanwhile, the current of the tip is recorded at a sampling rate of 20 kHz. Thousands of tip current-displacement curves are collected to construct the conductance histogram without data selection. The conductance measurement is carried out at a bias voltage of 50 mV.

The procedure of the $I$–$t$ tests with STM feedback loop turned off is briefly described as follows: First, the STM tip is stabilized at a fixed distance set according to the tunneling parameters ($I_t$ = 0.8 nA and $V_{bias}$ = 50 mV). Then we turn off the STM feedback loop and record the tip current for 20 s. When no molecule bridges the two electrodes, there is only the base current. Due to thermal, mechanical drift or molecular motions, the tip can contact the molecules. Once a molecular junction is formed, the current suddenly increases, which means that one or more molecules bridge the two electrodes. The $I$–$t$ curves are recorded with STM feedback loop of 0 at the substrate potentials of 0 and −0.5 V.

The procedure of $I$–$V$ measurements is described as follows: Firstly, the STM tip is driven toward the substrate to a preset current value (50 nA) via piezoelectric control. Then we turn off the STM feedback loop, an external pulse voltage is applied on z-piezo to bring STM tip 1 nm into the substrate surface to ensure tip contact with the substrate. Next, the tip is withdrawn until the tip current value reaches

the single-molecule conductance value (Equal to the molecular conductance $10^{-3.0}$ $G_0$ of 4-MTBA multiplied by 50 mV bias). When the single-molecule conductance is detected, the tip will be fixed over the substrate. The tip current is recorded during the substrate potential is swept from 0 to −0.5 V.

## In situ Raman measurements

Raman experiments were carried out on a confocal microscope Raman system (Renishaw InVia). The excitation wavelength was 632.8 nm, and a ×50 microscope objective with a numerical aperture of 0.55 was used in all Raman measurements. In situ electrochemical Raman experiments were carried out in a homemade Raman cell with potential control at an CHI-660E potentiostat. The 120 nm Au @ ca. 2 nm $SiO_2$ nanoparticles synthesized following previous reports[55], dropped on the Au(111) electrode as Raman-signal amplifiers.

## Computational methods

All the theoretical calculations were performed by Vienna Ab Initio Simulation Package (VASP)[56,57] software based on the density function theory (DFT). The effects between electron exchange and correlation were described by the Perdew–Burke–Ernzerhof (PBE) functional, which was one of the most well-established generalized gradient approximation (GGA)[58]. The interactions between electrons and ions were represented by projected augmented wave (PAW)[59,60] potentials with a cutoff energy of 450 eV. The Brillouin-zone integration[61] was approximated by $1 \times 2 \times 1$ k-point sampling grid. The convergence criterion for the energy calculation was set to $1.0 \times 10^{-4}$ eV, while the force tolerance of structure optimization was set to 0.01 eV Å$^{-1}$. The van der Waals interaction was described by DFT-D3 method[62]. To verify the impact of the model, a series of Au models were constructed by DFT calculation. A lattice cell of a three-layer $2 \times 2$ Au(111) surface and a three-layer $4 \times 2$ Au(111) surface with more than 15 Å vacuum layer were applied to prove the tiny influence of model size on the trend of adsorption energy. In order to approach the experimental environments, another three-layer $2 \times 2$ Au(111) surface with a thinner vacuum layer was constructed to simulate the presence of gold electrodes at both terminals. The results showed that the thickness of the vacuum layer also has barely effect on the trend of adsorption energy for this system. Furthermore, the three-layer $(2 \times 2)$ Au(111) surface under 25 $H_2O$ molecules environment with or without $Na^+$ were used to optimize the adsorption structure of 4-MTBA molecule by AIMD simulations. The AIMD simulations were performed in the canonical ensemble (NVT)[63,64] with the Nose-Hoover thermostat at 300 K for a time period of 5 ps with a time step of 0.5 fs. Then the 4-MTBA was removed from the equilibrated 4-MTBA/aqueous phase/Au(111) system for constructing aqueous phase/Au(111). And a standard DFT calculation was used to optimize the 4-MTBA/aqueous phase/Au(111) and aqueous phase/Au(111) system.

## Data availability

The data that support the findings of this study are available within the article, its Supplementary Information files. All data underlying the findings of this work are available from the corresponding author upon request.

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

## Acknowledgements

X.-S.Z., Y.-H.W., and Y.S. acknowledge financial support from the National Natural Science Foundation of China (nos. 22102150, 22172146, 21872126, and 21573198), the Zhejiang Provincial Natural Science Foundation of China (no. LQ21B030010) and the Leading Talent Program of Science and Technology Innovation in Zhejiang (no. 2020R52022).

## Author contributions

X.-S.Z. and Y.-H.W. conceived and designed the experiments. X.-S.Z., L.T., and J.-F.Z. constructed the ECSTM-BJ setup and performed single-molecule conductance measurements. Z.Y. and X.-C.L. performed nanoparticles synthesis and Raman and experiments, the theoretical analysis was coordinated by Y.-J.G. Finally, Y.S., Y.-H.W., and X.-S.Z. wrote the manuscript with input from all co-authors.

## Competing interests

The authors declare no competing interests.
