## [Peer Review File · Nature Communications]

REVIEWER COMMENTS

Reviewer #1 (Remarks to the Author):

In this manuscript, Zhou and co-workers demonstrated a new strategy based on the electrified localized metal cations tuning to achieve a reversible single-molecule switch with conductance On/Off high ratio using an electrochemical scanning tunneling break junction technique. The single-molecule conductance measurements of aliphatic and aromatic carboxylic acids were performed in the electrolyte solution containing metal cations to study the interfacial Au-carboxyl contacts. Combined with in situ Raman technique of SHINERS, interfacial molecular evidence clearly demonstrated a switching mechanism arising from the influence of localized cations on Au-carboxyl contacts. This work is interesting and noteworthy because it addresses a problem for molecular-metal contacts. It is the reviewer's opinion that this manuscript is suitable for a publication in Nat. Commun.

Here are some comments.

1. In Fig.2d, the authors describe the step stretching displacement (Δz) distribution with a Gaussian fit, but no corresponding histogram is observed. It should be at least added in the supplementary information.
2. It is interestingly found that the local concentrated metal cations in OHP can significantly affect the Au-COO⁻ contacts at lower potentials than PZC, thereby hindering the formation of molecular junctions. In this case, the concentrations of metal cations in solution might also have an impact. I suggest the authors add potential-dependent single-molecule conductance measurements in another concentration of metal cations to confirm this interesting phenomenon.
3. 4-MTBA has an asymmetric anchoring group. It has been reported that the asymmetric molecular structure has an effect on electron transport when changing the polarity of the bias voltage. Can these locally cation-tuned switches work when changing the polarity of the bias voltage?
4. Fig.3a shows the CVs of Au (111) in 50 mM NaClO₄ solution. And authors compared their single-molecule experimental results with that in 50 mM HClO₄. Thus, the CVs in HClO₄ solution should be provided to confirm their similar electrochemical behaviors.
5. There are some typo and format errors throughout the manuscript. Please check them.

Reviewer #2 (Remarks to the Author):

This paper reports on the effect of local cations in the electric double layer on the formation of single molecule junctions (SMJs). The used molecules bear carboxylate groups which are affected by a) the potential applied to the substrate of the device, b) the used electrolyte.

The subject is of interest for the readers of nature communication but in my opinion, the claim that a reversible single molecule switch is observed is strongly oversold and the highlighted On/Off ratio exceeding 10⁴ is not correct. Because of these two reasons, I cannot recommend this paper for publication in nature communication.

I agree that the authors show that molecules bearing terminal COOH groups do not bind to an STM tip in a similar way when the potential applied to the Au(111) substrates is varied. When it is higher than the potential of zero charge in NaClO₄ solution, SMJs with conductance around 10⁻³ G₀ can be generated. However, when the applied potential is lower than the potential of zero charge, the authors show that it is now impossible to fabricate an SMJ. Changing NaClO₄ solution to NH₄Cl or HClO₄ solutions makes it possible to generate SMs at various applied potential.

This is an interesting observation that has interest for the researchers working in the field of SMJs and I agree that it reveals that the local metal cations in the Helmholtz plane can significantly affect Au-carboxyl contacts to form molecular junctions for electron transport.

However, I also believe that no switching effects is really demonstrated in this paper. Indeed, to demonstrate a switch, it is in my opinion necessary to show I(V) curves (at constant bias) of the single molecule junctions and not only the fact that the applied potential to the substrate has an impact on the number of SMJ that can be obtained. V gate could be swept and during the sweep

breaking of the SMJ could be observed. This is not what is reported in this manuscript. This is particularly true in the I-t curves presented in Figure 5 (d) in which no switch between On and Off states are observed. The red figure is measured with a applied potential of 0 V and a "stable" (8 seconds) SMJ is obtained while the green curve is measured with an applied potential of -0.5 V and No SMJ are seen during 8 seconds. Besides the way the I(t) curves are obtained with Au tip driven to approach until reaching a current value of 4 nA via piezo electric control with a bias voltage of 50 mV seems to be impossible to use when the gate voltage applied to the substrate is -0.5 V as in these conditions no SMJs are generated. In other words, the distance between the tip and the substrate for the green curve is difficult to control and if it is too high it is likely that the observed low current (at the limit of detection) is not significant of the claimed SMJ switch. This clearly question the reported On/OFF ratio above 104.

Note also that such a apparent "switch", with a similar On/OFF ratio,, can be obtained by just retracting the tip in STM-BJ experiments which is what is observed in one of the red curves of figure 2b.

Note also that I(V_{gate}) curves at fixed bias have been reported by several groups working with junctions using few electroactive molecules or few conjugated polymer wires and that switch behavior when changing the gate voltage was in these studies been demonstrated (see Tao or Lacroix work on Polyaniline for instance or Amatore or Mayor and Borguet work on small electroactive molecules) (I agree that ON/OFF ratios are small but switching behavior is clearly demonstrated)

Overall, I believe that the results reported in this manuscript are interesting but that the way they are presented highlighting a switching behavior is misleading and that a reversible single molecule switch with On/Off ratio exceeding 104 is not demonstrated.

There are also few technical issues to solve prior to publication in another journal. Indeed, the used reference electrode (Platinum??) is not an usual reference electrode in electrochemistry and its potential is likely to vary with the used solution (it just does not act as a reference). Moreover, from the experimental section it is not easy to see if the same platinum electrode acts as reference and counter electrode or if the used setup is using four electrodes (a Pt reference, a Pt counter, the Au(111) substrate) and the Au STM tip) This must be better explained. Finally, I am not so sure about the attribution of the electrochemical signal observed in Figure 3a mainly because the used reference is a Pt tip. It is attributed to deprotonation or protonation of the carboxyl groups, but this is NOT a redox reaction and carboxyl group are not electroactive (or is they are, at high potential, decarboxylation can occur) so deprotonation and protonation alone cannot be responsible of the observed current in figure 3a. This clearly need to be clarified using a more classical reference electrode prior to any publication.

Reviewer #3 (Remarks to the Author):

The study by Tong, Yu, Gao et al reports single-molecule conductance measurements under electrochemical control. The conductance depends on both the potential of the electrode, and the presence of metal ions in solution, in a way that is exploited by the authors to demonstrate switching behaviour. The conclusions drawn from the conductance measurements are supported by Raman spectroscopy and control experiments. I find the study well carried out, interesting, and suitable for Nature Communications. There are some points, however, I would like to see clarified before publication.

How are the molecules self-assembled on the Au surface? Are factors such as density and order important?

I have some questions about the DFT part. Firstly, is it appropriate for the problem at hand? Without benchmarking the functional and basis set, modelling the second gold surface and the solvent environment, and accounting for effects such as basis set superposition error, how quantitative do the authors expect the calculated adsorption values to be? If the trend is only expected to be qualitative, then this should be stated.

The authors state in the introduction that 'DFT simulations reveal strong molecular carboxyl-metal-

cation coordination at the negatively charged electrode surface'. I'm not sure that the calculated adsorption energies show the strength of the carboxyl-metal cation coordination, but rather the effect of the additional presence of the gold cluster in the specific geometries presented.

Does adding in NH₄Cl or HClO₄ (or indeed any of the salts) alter the pH of the solution, and in turn does this affect the conductance (e.g. Ref 33- J. Phys. Chem. Lett. 2020, 11, 23, 10023–10028)? The peak at -0.5 V for HClO₄ is noticeably higher conductance than for the others. Could the authors add a table of conductances (including errors) to the supplementary information?

The peak for the NH₄Cl at 0 V is particularly weak. Are the authors sure there is a peak really there, and is the weakness also an effect of pH?

It is surprising that the switching effect is reversible with Ca²⁺, previous reports have shown that binding leads to the loss of the reversible features in the CV, see e.g. *Electrochimica Acta* 53 (2008) 6759–6767. Could the authors comment on this?

Why is the aliphatic MPA more conductive than the aromatic MTBA? Pi-conjugated molecules normally have closer levels for mediating transport. Is this just an effect of molecular length or the molecular structure affecting the pK_a of the acid?

ON/OFF ratio is just one part of a switch performance. How many cycles can be achieved? What are the limitations on frequency?

Here are some minor comments.

There are four typos in the second sentence of the abstract.

ΔE, written in Fig. 1b, should be defined in the caption or rewritten as E_{ads}.

Line 43. 'Wielydy' should read 'widely'

Line 113. Full stop missing.

Line 121. The authors state that 'Experimental details could be found in the Supplementary Information', however most are in the Methods section.

In Fig 4 (and Fig 1.) The authors use <<PZC, ~PZC, >>PZC, these inequalities should have a value on either side (e.g. V << PZC).

In Fig. 5d, put a y-axis label on at 0 nA so the magnitude of the current (not just the variation) can be seen.

Line 272 'Quantitative statistics' seems a bit of an exaggeration of fitting to a Gaussian to a section of the I-t trace. Why just choose a section? I think at least add some error bars to value of 3.8 nA.

Line 278. 'To conclusion' should read 'In conclusion'

Replies to Reviewers

Reviewer: 1

Comments to the Author

In this manuscript, Zhou and co-workers demonstrated a new strategy based on the electrified localized metal cations tuning to achieve a reversible single-molecule switch with conductance On/Off high ratio using an electrochemical scanning tunneling break junction technique. The single-molecule conductance measurements of aliphatic and aromatic carboxylic acids were performed in the electrolyte solution containing metal cations to study the interfacial Au-carboxyl contacts. Combined with in situ Raman technique of SHINERS, interfacial molecular evidence clearly demonstrated a switching mechanism arising from the influence of localized cations on Au-carboxyl contacts. This work is interesting and noteworthy because it addresses a problem for molecular-metal contacts. It is the reviewer's opinion that this manuscript is suitable for a publication in Nat. Commun.

Response: We greatly appreciate the reviewer for his/her high appraisal of our work.

1) In Fig.2d, the authors describe the step stretching displacement (Δz) distribution with a Gaussian fit, but no corresponding histogram is observed. It should be at least added in the supplementary information.

Response: As shown in Fig. R1, the step stretching distance (Δz) distribution can be Gaussian fitted into a main peak at 0.42 nm. By adding the snapback distance of breaking Au–Au contacts to the relative displacement Δz , it can be found that the most probable absolute displacement for the stretching process is 0.92 nm, which is comparable to the length of isolated 4-MTBA optimized by DFT/ B3LYP method with 6-311+G (d, p) basis sets via the Gaussian 09 software package. This confirms the formation of single-molecule junctions.

Fig. R1. Gaussian fitting of displacement distance (Δz) distribution

According to reviewer's suggestion, we have revised the discussion to “*the step stretching displacement (Δz) distribution with a Gaussian fitted peak at 0.4 nm (Supplementary Fig.5)*” in the revised manuscript, and supplemented Fig. R1 in the revised supplementary information.

2) It is interestingly found that the local concentrated metal cations in OHP can significantly affect the Au-COO⁻ contacts at lower potentials than PZC, thereby hindering the formation of molecular junctions. In this case, the concentrations of metal cations in solution might also have an impact. I suggest the authors add potential-dependent single-molecule conductance measurements in another concentration of metal cations to confirm this interesting phenomenon.

Response: We have supplemented the electrochemical gating of single-molecule conductance measurements in 0.1 mM 4-MTBA + 1 M Ca(ClO₄)₂ solution. The potential-dependent 1D conductance histogram of 4-MTBA in 1 M Ca(ClO₄)₂ are shown in Fig. R2. As the potential decreases, the conductance peak at $10^{-3.0} G_0$ becomes weaker and disappears below -0.2 V in 1 M Ca(ClO₄)₂, which is 0.1 V earlier than in 0.05 M Ca(ClO₄)₂. This suggests that higher concentration of cations in the bulk solution leads to more concentrated metal cations in OHP affecting the Au-COO⁻ contact.

Fig. R2 The potential dependent 1D conductance histogram of 4-MTBA in 1 M $\text{Ca}(\text{ClO}_4)_2$. The counts are normalized by the numbers of conductance curves used. All potentials are specified with respect to Pt.

In response to reviewer's concerns, we have added a discussion "*Increasing the electrolyte concentration can make the conductance peak disappear at higher potentials (Supplementary Fig.11)*" in the revised manuscript, and added Fig.R2 in the revised supplementary information.

3) 4-MTBA has an asymmetric anchoring group. It has been reported that the asymmetric molecular structure has an effect on electron transport when changing the polarity of the bias voltage. Can these locally cation-tuned switches work when changing the polarity of the bias voltage?

Response: The cation-tuned conductance switch remains effective when changing the

polarity of the bias voltage. We have supplemented the single-conductance measurements in 50 mM NaClO₄ solution with a fixed negative bias voltage of -50 mV ($E_{\text{substrate}} - E_{\text{tip}}$). As shown in Fig. R3, an obvious conductance peak at $10^{-2.9} G_0$ can be observed at 0 V, while it disappears at -0.5 V, similar to that with a positive bias voltage of 50 mV in Fig. 2c. Therefore, the potential-controlled conductance switch can still work when changing the polarity of the bias voltage.

Fig. R3. 1D conductance histograms of 4-MTBA obtained in 50 mM NaClO₄ solution at 0 V and -0.5 V with a bias voltage of -50 mV. The counts are normalized by the numbers of conductance curves used. All potentials are specified with respect to Pt.

In light of the reviewer's comments, we have added "*In addition, the cation-tuned conductance switch remains effective when changing the polarity of the bias voltage (Supplementary Fig.7)*" in the revised manuscript, and Fig. R3 in the revised supplementary information.

4) Fig.3a shows the CVs of Au (111) in 50 mM NaClO₄ solution. And authors compared their single-molecule experimental results with that in 50 mM HClO₄. Thus, the CVs in HClO₄ solution should be provided to confirm their similar electrochemical behaviors.

Response: We have supplemented the CVs of Au (111) in 50 mM HClO₄ solution. As shown in Fig. R4, there is one pair of well-defined reversible peaks at about 0.05 V. A linear correlation between current density of oxidation (red square) or reduction (blue square) with the scan rates is found. This proves that the reversible peaks arise from the 4-MTBA assembled on the Au(111) interface, which is similar to the 4-MTBA assembled on the Au(111) in 50 mM NaClO₄ solution.

Fig. R4. (a) CVs of Au(111) obtained in 0.1 mM 4-MTBA + 50 mM HClO₄ solution with different scan rates. (b) The plot of the current density of the anodic (red square) and cathodic (blue circle) peaks against the scan rates. All potentials are specified with respect to Pt.

According to reviewer's suggestion, we have added a discussion "*In addition, such reversible peaks can also be observed in the CVs using SCE and Pt as the reference electrodes in 50 mM NaClO₄ and 50 mM HClO₄ solutions (Supplementary Fig. 9), respectively.*" in the revised manuscript, and supplemented Fig. R4 in the revised supplementary information.

5) There are some typo and format errors throughout the manuscript. Please check them.

Response: We thank the reviewer very much for the correction, and have carefully checked the manuscript and revised typo and format errors marked with yellow.

Reviewer: 2

Comments to the Author

This paper reports on the effect of local cations in the electric double layer on the formation of single molecule junctions (SMJs). The used molecules bear carboxylate groups which are affected by a) the potential applied to the substrate of the device, b) the used electrolyte.

The subject is of interest for the readers of nature communication but in my opinion, the claim that a reversible single molecule switch is observed is strongly oversold and the highlighted On/Off ratio exceeding 10^4 is not correct. Because of these two reasons, I cannot recommend this paper for publication in nature communication.

Response: We greatly appreciate the reviewer's constructive comments. To further prove the conductance On/Off ratio, we have supplemented the I-V (at a constant bias) tests. As shown in Fig. R5, obviously, the current jumps to a low value or a high value during a negative or positive potential sweep, respectively. These be attributed to the breakdown and formation of molecular junctions. The relative current difference is about 3.83 nA comparable to single-molecule conductance in the break junction measurements. When the gate potential is lower than -0.4 V, tip current is turned off for all I-V parallel tests (Fig. R5), which is consistent with the disappearance of conductance peak below -0.4 V in single-molecule break junction experiments. These further prove the local-cation controlled single-molecule switch.

In light of reviewer's constructive comments, we have changed the title to "*Local cation-tuned reversible single-molecule switch in electric double layer*" and removed the highlight of On/Off ratio exceeding 10^4 in the abstract. We have added Fig. R5 and a discussion "*To further test the switching effects, the I-V measurements have been carried out in 0.1 mM 4-MTBA + 50 mM NaClO₄ solution with a constant bias of 50 mV. The potentials of Au(111) substrate are swept between 0 to -0.5 V. Fig. 5e shows a typical I-V curves recorded upon formation of molecular junctions. Obviously, the current jumps to a low value or a high value during a negative or positive potential sweep, respectively. These can be attributed to the breakdown and formation of molecular junctions. The relative current difference is about 3.83 nA comparable to single-molecule conductance in the break junction measurements. When the gate*

potential is lower than -0.41 V, tip current is turned off for all I-V parallel tests (Supplementary Fig.19), which is consistent with the disappearance of conductance peak below -0.4 V in single-molecule break junction experiments. These further prove the local-cation controlled single-molecule switch.” in the revised in the manuscript.

I agree that the authors show that molecules bearing terminal COOH groups do not bind to an STM tip in a similar way when the potential applied to the Au(111) substrates is varied. When it is higher than the potential of zero charge in NaClO₄ solution, SMJs with conductance around $10^{-3} G_0$ can be generated. However, when the applied potential is lower than the potential of zero charge, the authors show that it is now impossible to fabricate an SMJ. Changing NaClO₄ solution to NH₄Cl or HClO₄ solutions makes it possible to generate SMs at various applied potential.

This is an interesting observation that has interest for the researchers working in the field of SMJs and I agree that it reveals that the local metal cations in the Helmholtz plane can significantly affect Au-carboxyl contacts to form molecular junctions for electron transport.

Response: We greatly appreciate the reviewer for his/her positive comments on our work.

1) However, I also believe that no switching effects is really demonstrated in this paper. Indeed, to demonstrate a switch, it is in my opinion necessary to show I(V) curves (at constant bias) of the single molecule junctions and not only the fact that the applied potential to the substrate has an impact on the number of SMJ that can be obtained. V gate could be swept and during the sweep breaking of the SMJ could be observed. This is not what is reported in this manuscript. This is particularly true in the I-t curves presented in Figure 5 (d) in which no switch between On and Off states are observed. The red figure is measured with an applied potential of 0 V and a “stable” (8 seconds) SMJ is obtained while the green curve is measured with an applied potential of -0.5 V and No SMJ are seen during 8 seconds. Besides the way the I(t) curves are obtained with Au tip driven to approach until reaching a current value of 4 nA via piezo

electric control with a bias voltage of 50 mV seems to be impossible to use when the gate voltage applied to the substrate is -0.5 V as in these conditions no SMJs are generated. In other words, the distance between the tip and the substrate for the green curve is difficult to control and if it is too high it is likely that the observed low current (at the limit of detection) is not significant of the claimed SMJ switch. This clearly questions the reported On/OFF ratio above 10^4 .

Response: To further test the switching effects, we have supplemented the I-V measurements with a constant bias of 50 mV in 0.1 mM 4-MTBA + 50 mM NaClO₄ solution. The procedure of I-V measurements is described as follows: First, the STM tip is driven toward the substrate to a preset current value (50 nA) via piezoelectric control. Then an external pulse voltage is applied on z-piezo to bring STM tip 1 nm into the substrate surface to ensure tip contact with the substrate. Next, the tip is withdrawn until the tip current value reaches single molecule conductance value (Equal to the molecular conductance $10^{-3.0} G_0$ of 4-MTBA multiplied by 50 mV bias). When the single-molecule conductance is detected, the tip will be fixed over the substrate, and the substrate potential is swept between 0 to -0.5 V. Fig. R5a the traces of I-V curves recorded upon formation of molecular junctions. Obviously, the current jumps to a low value or a high value during a negative or positive potential sweep, respectively. These can be attributed to the breakdown and formation of molecular junctions. The relative current difference is about 3.83 nA comparable to single-molecule conductance in the break junction measurements. When the gate potential is lower than -0.41 V, tip current is turned off for all I-V parallel tests (Fig. R5b), which is consistent with the disappearance of conductance peak below -0.4 V in single-molecule break junction experiments. These further prove the local-cation controlled single-molecule switch.

For I-t measurements, we apologize for the wrong description of the experimental details. We agree that the tip-to-substrate distance is difficult to know in STM-BJ experiments. Certainly, when tip-to-substrate distance is larger than the molecular length, molecular junction will not form to observe the current spikes in the I-t curves. However, in the constant current mode of STM, the distance from the tip to the substrate can be controlled by the preset current value. In our experiments, we optimized the

current setpoint values to 4 nA at the substrate potential of 0 V, at which the molecular junction can be formed based on the break junction measurements. We recorded the I-t curves with a very low feedback loop of 0.01 at the substrate potentials of 0 and -0.5 V, respectively. It was found that the current spikes ascribed to form molecular junctions disappeared at -0.5 V, which verified this local-cation control switch consistent with the results of break junction measurements.

Fig. R5. (a) A typical I-V curve in 0.1 mM 4-MTBA + 50 mM NaClO₄ with a constant bias of 50 mV. (b) I-V parallel tests obtained by simultaneously sweeping the tip and substrate potential from 0 to -0.5 V.

According to reviewer's suggestion, we have added Fig. R5 and a discussion "To further test the switching effects, the I-V measurements have been carried out in 0.1 mM 4-MTBA + 50 mM NaClO₄ solution with a constant bias of 50 mV. The potential of Au(111) substrate is swept between 0 to -0.5 V. Fig. 5e shows I-V curves recorded upon formation of molecular junctions. Obviously, the current jumps to a low value or a high value during a negative or positive potential sweep, respectively. These can be attributed to the breakdown and formation of molecular junctions. The relative current difference is about 3.83 nA comparable to single-molecule conductance in the break junction measurements. When the gate potential is lower than -0.41 V, tip current is turned off for all I-V parallel tests (Supplementary Fig.19), which is consistent with the disappearance of conductance peak below -0.4 V in single-molecule break junction experiments. These further prove the local-cation controlled single-molecule switch."

in the revised in the maintext. We have changed the title to “*Local cation-tuned reversible single-molecule switch in electric double layer*” and removed the highlight of the On/Off ratio in the Abstract. We have supplemented the experimental details of I-V and I-t tests in Methods section.

Note also that such an apparent “switch”, with a similar On/OFF ratio, can be obtained by just retracting the tip in STM-BJ experiments which is what is observed in one of the red curves of figure 2b.

Response: We agree that molecular junctions would be broken by withdrawing the tip away from the substrate beyond the molecular length, which can cause a rapid decrease in conductance for all kinds of molecular junctions. Such mechanical control has also been used to realize conductance switching in a single-molecule junction (*Nat. Nanotech.*, 2009, 4, 230–234; *Nat. Nanotech.* 2012, 7,35–40; *Angew. Chem. Int. Ed.*,2023, doi: 10.1002/ange.202302693). In our work, we found that the electrified localized metal cations in outer Helmholtz plane can strongly coordinate with molecular carboxyl-groups at the negatively charged electrode surface, which hinders the formation of molecular junctions for electron transport. Consequently, the step features in the conductance traces disappear at -0.5 V. This can also realize single-molecule conductance switch by potential control in our work instead of retracting the tip.

Note also that I(V_{gate}) curves at fixed bias have been reported by several groups working with junctions using few electroactive molecules or few conjugated polymer wires and that switch behavior when changing the gate voltage was in these studies been demonstrated (see Tao or Lacroix work on Polyaniline for instance or Amatore or Mayor and Borguet work on small electroactive molecules) (I agree that ON/OFF ratios are small but switching behavior is clearly demonstrated)

Response: We have cited the papers (*Anal. Chem.* 2011, 83, 9709–9714; *J. Am. Chem. Soc.* 2005, 127, 9235-9240; *J. Am. Chem. Soc.* 2020, 142, 7732-7736) that use I-V characteristics to show the current on/off ratio of molecular junctions, and supplemented the I-V tests (Fig.R5) to confirm the local cation-controlled switch

behavior.

Overall, I believe that the results reported in this manuscript are interesting but that the way they are presented highlighting a switching behavior is misleading and that a reversible single molecule switch with On/Off ratio exceeding 10^4 is not demonstrated.

Response: We greatly appreciate the reviewer for his/her constructive comments, and we have further proven such switching behavior using I-V tests, and have changed the title to “*Local cation-tuned reversible single-molecule switch in electric double layer*” and removed the highlight of On/Off ratio exceeding 10^4 in the revised Abstract.

2) There are also few technical issues to solve prior to publication in another journal. Indeed, the used reference electrode (Platinum??) is not an usual reference electrode in electrochemistry and its potential is likely to vary with the used solution (it just does not act as a reference). Moreover, from the experimental section it is not easy to see if the same platinum electrode acts as reference and counter electrode or if the used setup is using four electrodes (a Pt reference, a Pt counter, the Au(111) substrate and the Au STM tip) This must be better explained. Finally, I am not so sure about the attribution of the electrochemical signal observed in Figure 3a mainly because the used reference is a Pt tip. It is attributed to deprotonation or protonation of the carboxyl groups, but this is NOT a redox reaction and carboxyl group are not electroactive (or is they are, at high potential, decarboxylation can occur) so deprotonation and protonation alone cannot be responsible of the observed current in figure 3a. This clearly need to be clarified using a more classical reference electrode prior to any publication.

Response: In our STM-BJ setup, a four-electrode electrochemical system was used, the same Pt ring and Pt wire were used as counter electrode and reference electrode in the different experiments, respectively. There are two main reasons for using Pt as a quasi-reference electrode in our experiments. One is that the home-made cell (about 200 μ L) for electrochemical STM experiments is too small to put in the commercial reference electrode. So most electrochemical STM experiments use metal wires as quasi-reference electrodes including Pt (*Nat. Catal.* 2021, 4, 850–859; *Nat. Mater.* 2019, 18,

357–363; *J. Am. Chem. Soc.* 2022, 144, 20126–20133). The other is the presence of chloride ions in the commercial reference electrodes of SCE and Ag/AgCl. Halogen ions can specifically adsorb and even etch the Au surface, especially at high potentials > PZC.

We have also supplemented the CVs of Au(111) in 0.1 mM 4-MTBA +50 mM NaClO₄ solution with different scan rates using SCE as the reference electrode. As shown in Fig. R6, there is one pair of well-defined reversible peaks at about 0.4 V vs. SCE. Plotting current density of oxidation (red square) or reduction (blue circle) with the scan rate, a linear correlation is also observed. The total charges of oxidation peaks are quantitatively estimated at about 14.5 μC/cm² consistent with the results using Pt as quasi-reference electrode in Fig. 3a. Such reversible peaks have also been observed at self-assembled monolayers of carboxylic acid molecules in previous reports (*Langmuir*, 2006, 22, 4420–4428; *Electrochim. Acta*, 2008, 53, 6759–6767), which are assigned to deprotonation or protonation of the carboxyl groups in positive or negative scans of the electrochemical potentials.

Fig. R6. CVs of Au(111) obtained in 0.1 mM 4-MTBA + 50 mM NaClO₄ solution with different scan rates. (b) The plot of the current density of the anodic (red square) and cathodic (blue circle) peaks against the scan rates. All potentials are specified with respect to SCE.

In light of the reviewer’s comment, we have added a statement “*In addition, such reversible peaks can also be observed in the CVs using SCE and Pt as the reference electrodes in 50 mM NaClO₄ and 50 mM HClO₄ solutions (Supplementary Fig. 9), respectively.*” in the revised maintext and the experimental detail “*In a four-electrode*

electrochemical system, the same Pt ring and Pt wire were used as counter electrode and reference electrode in all electrochemical STM-BJ experiments, respectively. The naturally formed (111) facet at an Au bead substrate and mechanically-cut Au STM tip (0.25 mm diameter, 99.999%, Alfa Aesar) were used as work electrodes.” in the revised Method section.

Reviewer #3 (Remarks to the Author):

The study by Tong, Yu, Gao et al reports single-molecule conductance measurements under electrochemical control. The conductance depends on both the potential of the electrode, and the presence of metal ions in solution, in a way that is exploited by the authors to demonstrate switching behavior. The conclusions drawn from the conductance measurements are supported by Raman spectroscopy and control experiments. I find the study well carried out, interesting, and suitable for Nature Communications. There are some points, however, I would like to see clarified before publication.

Response: We greatly appreciate the reviewer for his/her positive and constructive comments of our work.

1) How are the molecules self-assembled on the Au surface? Are factors such as density and order important?

Response: The single-molecule conductance measurements were performed in a solution containing 0.1 mM 4-MTBA. The molecules can adsorb on Au(111) through the anchor groups of methyl sulfide (SMe) to form self-assembled monolayer. We also immersed Au(111) in 0.1 mM 4-MTBA aqueous solution for 10 min. We then took it out and dried it under a nitrogen purge. The break junction measurements display an obvious conductance peak at around $10^{-3.0} G_0$ in the conductance histograms (Fig. R7), consistent with the result of conductance measurements in the solution.

The degree of order of the molecular assembly layer should might have no effect on the formation of molecular connections. Fig. R7a shows the STM images of Au(111)

in the 0.1 mM 4-MTBA aqueous solution. No ordered structure of the SAM of 4-MTBA was found under our experimental conditions. According to the previous report (*Angew. Chem. Int. Ed.* 2019, 58, 14534–14538), the formation probability of methyl sulfide-linked molecular junction as a function of molecular concentration from 1×10^{-7} to 5×10^{-5} M fits well with the Langmuir isotherm. Continuing to increase the concentration of molecules, the intensity of the conductance peak hardly changes due to molecular adsorption saturation. Thus, the molecular concentration of 0.1 mM (1×10^{-4} M) can ensure that there is enough molecular adsorption to maximize the probability of junction formation.

Fig. R7. (a) STM images of self-assembled 4-MTBA monolayer on the Au(111) substrate. Imaging conditions: $E_{\text{bias}} = 50$ mV, $I_{\text{tip}} = 0.5$ nA. (b) The 1D conductance histograms obtained at self-assembled 4-MTBA monolayer.

In response to reviewer’s concerns, we have added the experimental details “*The single-molecule conductance measurements were performed in a solution containing 0.1 mM 4-MTBA + 50 mM electrolytes. The molecules can adsorb on Au(111) through the anchor groups of methyl sulfide (SMe) to form self-assembled monolayer*” in the revised manuscript.

2) I have some questions about the DFT part. Firstly, is it appropriate for the problem at hand? Without benchmarking the functional and basis set, modelling the second gold surface and the solvent environment, and accounting for effects such as basis set superposition error, how quantitative do the authors expect the calculated adsorption

values to be? If the trend is only expected to be qualitative, then this should be stated. The authors state in the introduction that ‘DFT simulations reveal strong molecular carboxyl-metal-cation coordination at the negatively charged electrode surface’. I’m not sure that the calculated adsorption energies show the strength of the carboxyl-metal cation coordination, but rather the effect of the additional presence of the gold cluster in the specific geometries presented.

Response: We greatly appreciate the reviewer for his/her constructive comments. We have supplemented DFT and AIMD calculations by simulating the second gold surface and the solvent environment to account for cation effect on the molecule-metal interaction. All the theoretical calculations were performed by Vienna Ab Initio Simulation Package (VASP) software. The effects between electron exchange and correlation were described by the Perdew-Burke-Ernzerhof (PBE) functional, which was one of the most well-established the generalized gradient approximation (GGA). The interactions between electrons and ions were represented by projected augmented wave (PAW) potentials with a cutoff energy of 450 eV. The Brillouin-zone integration was approximated by $1\times 2\times 1$ k-point sampling grid. The convergence criterion for the energy calculation was set to 1.0×10^{-4} eV, while the force tolerance of structure optimization was set to 0.01 eV \AA^{-1} . The van der Waals (vdW) interaction was described by DFT-D3 method.

To verify the impact of the model, two series of Au models were constructed. As shown in Fig. R8, a lattice cell of three-layer 2×2 Au(111) surface and three-layer 4×2 Au(111) surface with more than 15 \AA vacuum layer were applied to prove the tiny influence of model size on the trend of adsorption energy. The DFT calculation results demonstrate that the adsorption of 4-MTBA with COOH (-0.60 eV) is significantly weakened than ones of 4-MTBA with COO⁻ (-2.11 eV) on the three-layer (2×2) Au(111) surface (48 Au atoms concluded). Certainly, the adsorption energy also exhibits the same trend on (4×2) Au(111) surface (96 Au atoms concluded), which provide solid evidence for the tiny effect of supercell size on adsorption energy.

Fig. R8. The optimized structures of adsorbed 4-MTBA with different forms of carboxyl groups on the different size of Au(111).

In order to approach the experimental environments, another three-layer 2×2 Au(111) surface with thinner vacuum layer was constructed to simulate the presence of gold electrodes at both terminals. The results showed that the thickness of the vacuum layer also has barely effect on the trend of adsorption energy for this system. Furthermore, the three-layer (2×2) Au(111) surface under 25 H₂O molecules environment with or without Na⁺ were used to optimize adsorption structure of 4-MTBA molecule by AIMD simulations. The AIMD simulations were performed in the canonical ensemble (NVT) with the Nose-Hoover thermostat at 300 K for a time period of 5 ps with a time step of 0.5 fs. Then the 4-MTBA was removed from the equilibrated 4-MTBA/aqueous phase/Au(111) system for constructing aqueous phase/Au(111). And a standard DFT calculation was used to optimize the 4-MTBA/aqueous phase/Au(111) and aqueous phase/Au(111) system.

Fig. R9 Energy fluctuations versus AIMD simulation time (a) 4-MTBA/25H₂O/Au(111) and (b) 4-MTBA/Na-25H₂O/Au(111) at 300 K.

Fig.R9a show the total energy against time for 4-MTBA/25H₂O/Au(111), and two optimized snapshots at 5.0 ps (green line) and 2.818 ps (orange line, lowest energy point), respectively. The geometric structures and energy of two snapshot are quite similar, indicating the structure of adsorbed 4-MTBA on the 25H₂O/Au(111) preserved well from 3 to 5 ps. Similarly, Fig.R9b show the total energy against time for 4-MTBA/Na-25H₂O/Au(111). The energy fluctuation is almost negligible from 1 to 5 ps,

indicating the structure have high thermodynamic stability. Therefore, the optimized snapshots of AIMD at 5.0 ps was applied in adsorption energy calculation.

Finally, we calculated the adsorption energy (E_{ad}) of 4-MTBA in different configurations by the following expression:

$$E_{ad} = E_{total} - E_{4-MTBA} - E_{complex}$$

Where E_{total} is the total energy of adsorbed 4-MTBA system, and E_{4-MTBA} is the energy of the 4-MTBA molecule in the gas phase, $E_{complex}$ is the energy of Au layers and aqueous phase under solution condition.

Fig. R10. The optimized structures of (b) deprotonated 4-MTBA with COO^- on Au(111), (c) protonated 4-MTBA with $COOH$ on Au(111), (d) deprotonated 4-MTBA with COO^- on $25H_2O/Au(111)$ and (e) deprotonated 4-MTBA with COO^- on $Na-25H_2O/Au(111)$.

Fig. R10 a and b demonstrate that the adsorption of protonated 4-MTBA with $COOH$ (-1.23 eV) is significantly weaker than that of deprotonated 4-MTBA with COO^- (-2.66 eV) on the three-layer (2×2) Au(111) surface. Similarly, the deprotonated 4-MTBA adsorb on the $25H_2O/Au(111)$ through $-COO^-$ group, whose O binding with surface Au strongly. When adding Na^+ , the hydrated metal cation make $-COO^-$ more than 4 Å from the Au surface on the $Na-25H_2O/Au(111)$, resulting in weakened adsorption. This supports the feasibility of controlling Au-carboxyl contact by local cations.

According to reviewer's suggestion, we have replaced Fig. 1b with Fig. R10, and revised the discussion to "To substantiate this hypothesis, we firstly performed density

function theory (DFT) and ab initio molecular dynamics (AIMD) calculations to analyse the interaction between carboxylic acid molecules and Au electrode in different configurations using the Vienna Ab Initio Simulation Package (VASP) software (see Methods and Supplementary Fig. 1 and 2 in Supplementary Information). Fig.1b-e shows the optimized structures of different configurations of 4-MTBA adsorbed on three-layer (2×2) Au(111) surface. The adsorption of protonated 4-MTBA with COOH ($E_{ads} = -1.23$ eV) is significantly weaker than that of deprotonated 4-MTBA with COO⁻ ($E_{ads} = -2.66$ eV). Similarly, the deprotonated 4-MTBA adsorb on the 25H₂O/Au(111) through -COO⁻ group, whose O binding with surface Au strongly. When adding Na⁺, the hydrated metal cation make -COO⁻ more than 4 Å from the Au surface on the Na-25H₂O/Au(111), resulting in weakened adsorption. These support the feasibility of controlling Au-carboxyl contact by local cations”, and changed the statement to “In situ Raman spectra reveal strong molecular carboxyl-metal-cation coordination at the negatively charged electrode surface” in the revised manuscript. The calculation methods and details have been added in the revised Methods section and Supplementary Information.

3) Does adding in NH₄Cl or HClO₄ (or indeed any of the salts) alter the pH of the solution, and in turn does this affect the conductance (e.g. Ref 33- J. Phys. Chem. Lett. 2020, 11, 23, 10023–10028)? The peak at -0.5 V for HClO₄ is noticeably higher conductance than for the others. Could the authors add a table of conductance (including errors) to the supplementary information?

Response: We have supplemented a table containing single-molecule conductance values and pH in different electrolyte solution. As shown in Table R1, the pH of solutions with a pH meter (Model PHS-3E pH Meter Manual, Shanghai INESA & Scientific Instrument Co.Ltd). The pH of 0.1 m M MTBA solutions containing NaClO₄, KClO₄, Mg(ClO₄)₂, Ca(ClO₄)₂ or NH₄Cl is close to 5 due to the dissociation equilibrium of -COOH \rightleftharpoons H⁺ + -COO⁻. The solution containing 50 mM HClO₄ becomes more acidic because HClO₄ is a strong acid. While the solution containing 50 mM NaF becomes less acidic, because NaF is a salt of a strong base (NaOH) and a weak acid

(HF) and undergoes a hydrolysis reaction $F^- + H_2O \rightleftharpoons HF + OH^-$. According to previous report (*J. Phys. Chem. Lett.*, 2020, 11, 23, 10023–10028), the conductance peak intensity in HClO₄ should be the weakest. However, our experiments found the opposite. Combining Raman spectroscopy and theoretical calculations, we prove that strong molecular carboxyl-metal-cation coordination at the negatively charged electrode surface hinders the formation of molecular junctions.

Table R1. Single-molecule conductance of 4-MTBA in different electrolyte solution and the pH of solution.

Electrolyte solution	pH	G at 0 V log(G/G ₀)	G at -0.5 V log(G/G ₀)
NaClO ₄	5.01±0.01	-3.00±0.15	-
KClO ₄	5.06±0.03	-3.02±0.10	-
Ca(ClO ₄) ₂	5.01±0.02	-2.75±0.22	-
Mg(ClO ₄) ₂	4.90±0.02	-2.92±0.11	-
NaF	6.56±0.01	-2.98±0.13	-
NH ₄ Cl	4.99±0.02	-2.73±0.14	-3.03±0.23
HClO ₄	1.77±0.02	-2.98±0.17	-3.06±0.21

According to reviewer’s suggestion, we have added description “*The Single-molecule conductance of 4-MTBA in different electrolyte solution and the pH of solution are summarized in Supplementary Table 1*” in the revised manuscript, added Table R1 in the revised supplementary information.

4) The peak for the NH₄Cl at 0 V is particularly weak. Are the authors sure there is a peak there, and is the weakness also an effect of pH?

Response: We have shown its 1D and 2D conductance histograms in Fig. R11a and b, it can be found a conductance peak and an obvious stretching states centered around $10^{-3.0} G_0$. There may be two main reasons for the weaker conductance peak intensity in NH₄Cl solution compared to other electrolyte solutions: (1) Hydrolysis reaction

$NH_4^+ + RCOO^- + H_2O \rightleftharpoons NH_3 \cdot H_2O + RCOOH$ promotes the protonation of the carboxylic acid, which decreases the junction formation probability through the deprotonated $-COO^-$ groups. (2) The strong specific adsorption of chloride ions on gold surface at potential $E > PZC$, have been proven a significant decrease in stability of gold atomic contacts (*Phys Rev B*, 1998, 58,6775; *Top. Curr. Chem.*, 2012, 313, 121–188), thereby affecting the stability of molecular junctions. This can be confirmed by the shorter step stretching displacement (Δz) of 0.33 nm in Fig. R11c compared to that 0.42nm in $NaClO_4$ solution.

Fig. R11. (a) 1D and (b) 2D conductance histograms of 4-MTBA in NH_4Cl solution. (c) Gaussian fitting of displacement distance (Δz) distribution

In response to reviewer’s concerns, we have added a discussion “*The weaker conductance peak at 0 V in NH_4Cl solution might arise from the strong specific adsorption of halide ions and hydrolysis reaction (Supplementary Fig.6).*” in the revised manuscript, and added Fig. R11 and above-mentioned discussion in the revised supplementary information.

5) It is surprising that the switching effect is reversible with Ca^{2+} , previous reports have shown that binding leads to the loss of the reversible features in the CV, see e.g. *Electrochimica Acta* 53 (2008) 6759–6767. Could the authors comment on this?

Response: Compared to our results, the loss of the reversible features in the CV in the reference (*Electrochim. Acta*, 2008, 53, 6759-6767) mainly arises from adjusting the pH of the solution to 8.5. The alkaline environment deprotonates the molecule to

coordinate with Ca^{2+} ions. Also, the CV in ref was performed on the SAM of 4-MBA, whereas we performed it in a solution containing 0.1 mM molecule. This can lead to a change in the coverage of molecules on the electrode surface, resulting in changing peak current in CV.

We also supplemented the CV of the SAM of 4-MTBA in 50 mM $\text{Ca}(\text{ClO}_4)_2$ solution at pH 8.5. As shown in the Fig. R12a, there are no current peaks consistent with result of 4-MBA in the reference. While adding 0.1 mM 4-MTBA in the solution, the CV shows broad peaks in Fig. R12b. A linear correlation between current density of oxidation (red square) or reduction (blue square) with the scan rates is found in the insert, revealing that these reversible peaks arise from the 4-MTBA assembled on the Au(111) interface.

Fig. R12. CVs of (a) the SAM of 4-MTBA and (b) 0.1 mM 4-MTBA on Au (111) in 50 mM $\text{Ca}(\text{ClO}_4)_2$ solution at pH 8.5, insert is the plot of the current density of the anodic (red square) and cathodic (blue circle) peaks against the scan rates.

6) Why is the aliphatic MPA more conductive than the aromatic MTBA? π -conjugated molecules normally have closer levels for mediating transport. Is this just an effect of molecular length or the molecular structure affecting the pK_a of the acid? ON/OFF ratio is just one part of a switch performance. How many cycles can be achieved? What are

the limitations on frequency?

Response: We agree with that the π -conjugated molecules generally exhibit larger conductance values with smaller HOMO-LUMO gaps for electron transport. While the molecular length is another important factor determining the single-molecule conductance. Our previous report found that the pH of solution or the pK_a of carboxylic acid molecule affects the probability of molecular junction formation, but has little impact on its conductance value (*J. Phys. Chem. Lett.* 2020, 11, 10023–10028). Thus, we think that the aliphatic MPA is more conductive than aromatic MTBA due to its shorter molecular length.

We have supplemented the cyclic tests by repeatedly cyclically sweeping the potential between 0 and -0.5 V, followed by single-molecule conductance measurements. As shown in the Fig. R13, the conductance peaks can repeatedly appear at 0 V and disappear at -0.5 V over 50 cycles in NaClO₄ solution. This demonstrates the good stability of localized cation-tuned reversible single-molecule switches in the electric double layer. For switching frequency, this may be a weak point of our single-molecule switch. This may depend on the rate at which the applied potential changes the electric double layer structure. In our I-V tests, the switching frequency is estimated at about 0.025 Hz (switching in 40 s when changing the gate potentials at 10 mV/s).

Fig. R13. The 1D conductance histograms after repeated cycle scanning of potentials between 0 and -0.5 V in cycles of (a)1, (b)5, (c)10, (d)15, (e)20, and (f)50.

In light of reviewer’s constructive comments, we have added a discussion “*In addition, cycling tests were also performed by repeatedly cycling the potential scan followed by single-molecule conductance measurements, the conductance peaks can repeatedly appear at 0 V and disappear at -0.5 V over 50 cycles (Supplementary Fig.18).*” in the revised manuscript, and added Fig. R10 in the revised supplementary information.

7) Here are some minor comments.

There are four typos in the second sentence of the abstract.

E, written in Fig. 1b, should be defined in the caption or rewritten as E_{ads} .

Line 43. ‘Wieldy’ should read ‘widely’

Line 113. Full stop missing.

Line 278. ‘To conclusion’ should read ‘In conclusion’

Response: We thank the reviewer very much for the correction, and have carefully checked the manuscript and revised typo and format errors marked with yellow.

Line 121. The authors state that ‘Experimental details could be found in the Supplementary Information’, however most are in the Methods section.

Response: We have corrected the sentence to “*Experimental details could be found in Method section*”.

In Fig 4 (and Fig 1.) The authors use $\langle \rangle$ PZC, these inequalities should have a value on either side (e.g. $V \ll$ PZC).

Response: We have redrawn Fig.1a.

In Fig. 5d, put a y-axis label on at 0 nA so the magnitude of the current (not just the variation) can be seen.

Response: We have marked 0 nA for the y-axis in the revised Fig. 5d.

Line 272 ‘Quantitative statistics’ seems a bit of an exaggeration of fitting to a Gaussian to a section of the I-t trace. Why just choose a section? I think at least add some error bars to value of 3.8 nA.

Response: We apologize for our imprecise, and have corrected the current value to 3.68 ± 0.55 nA based on Gaussian fit in the entire I-t trace.

REVIEWER COMMENTS

Reviewer #1 (Remarks to the Author):

The authors addressed the questions raised by the reviewers and I suggest to accept it as it is.

Reviewer #2 (Remarks to the Author):

I read carefully the revised version of the manuscript now entitled « local Cation tuned reversible single-molecule switch in electric double layer. It is much more convincing and its subject is clearly of interest for the readers of Nature communication.

Many of my initial comments have been taken into account. The title has been changed and is now more in line with the findings and the I/V characterization in figure 5f now clearly show a switch between On and Off state. I thus recommend publication in nature communication. However, there is still some important modifications to address prior to publish the paper.

The three following points must be changed.

a) CV of figure 3a show a peak which is attributed to deprotonation/protonation of the carboxyl groups with the help of two references given (ref40 et 41). The author MUST add that there is no faradic processes taking place within this electrochemical potential range. This fact is clearly stated in Reference 41 and even though it is true that in 2008 a similar peak as the one shown here was reported (in reference 40) it remains important to say that this is not linked to any faradic processes. (deprotonation/protonation are not redox events) From reference 41 this is clearly stated the P1 peak is not attributed to protonation/deprotonation but to a reorganization of the ad layers on the surface.

b) I still have problem with the claimed ON/OFF ratio of conductance (or current) Indeed, the paper clearly demonstrate that there is a preferred conductance peak in some electrolyte when the applied potential is positive and that there is no conductance peak (i.e ; no single molecule junction) when the potential is below -0.3V (see figure 3b for instance) but this does not mean that the ON/OFF ratio exceeds $6.7 \cdot 10^4$ as claimed. Indeed, if we concentrate on the value of the measured current when no SMJs are obtained (at -0.5V), in figure 3b, there is current counts at all current values including for currents higher than that observed for the peak at 0.1 V. So the ratio of current is not $6.7 \cdot 10^4$ and can in fact be anything. If one now looks at the curves given in figure 5f, the current ON is around 4 nA and the current OFF is below 1 nA but is not at all $6.7 \cdot 10^4$ smaller than the current ON. Current Off is not small because there is leakage and in fact no SMJ are created but the tip is still not far away to the surface and if there is a little bit feedback loop as stated in the caption then the tip will try to move and get closer to the surface and reach the current set point used. In my opinion it is impossible to state that the ON/OFF ratio exceeds for order of magnitude as stated in Figure 5

c) My last problem is the I(t) curves in Figure 5d. Usually I(t) curves on SMJs are done without any feedback loop on and telegraphic style showing ON/OFF switch with time and (thus able to measure on/off ratio) can be obtained. Such curves must be done without feedback, See for two recent examples (Nano Letters 2021, 21 (15), 6540-6548 and J. Am. Chem. Soc. 2019, 141, 14788–14797 which could be cited). In Figure 5d a current set point of 4 nA is applied so it is impossible to have a current clearly different from 4 nA. Both curves (the red and the green) have been done with a current feedback thus i do not see exactly what they mean. The red curve is centered at 4 nA because of the set point and noisy so i do not exactly see what they mean.

Overall the paper is interesting with a very nice new concept but On/off ratio above 10

power 4 remains in my opinion not clearly demonstrated in the presented data.

Reviewer #3 (Remarks to the Author):

The study is improved by enacting the changes. I still find think it is suitable for Nature Communications, as the science is sound and the results are convincing and well-supported by the data, but some issues need to be clarified.

Regarding the switch performance I'm a little confused by Fig. 5. Is it correct that the conductance values in Fig. 5a – 5c are taken from fitting to conductance of many junctions followed by averaging? This should be clarified.

In Fig. 5d the $I(t)$ trace at -0.5 V has a mean value of $\sim 4\text{nA}$, so if translated into a Gaussian as done for the trace at 0 V in Fig. 5e it will appear at the same mean value but narrower, as there are no spikes present. I'm not sure what this adds to the switching behaviour, in my mind these $I(t)$ traces at a fixed bias imply the device could act as a sensor rather than a switch, - it reminds me of e.g.

<https://www.nature.com/articles/ncomms13868>.

Additionally, regarding the issue of reporting the ON/OFF ratio as 6.7×10^4 – I find removing it from just the abstract a strange course of action. Either it is correct, and can remain wherever, or it is incorrect/dubious, and then should be removed or heavily caveated.

Overall, some re-writing or restructuring of the discussion around Fig. 5 should be undertaken to clarify these, discussing limitations, and possibly other potential uses. Finally, there are still typos remaining, including some that I explicitly mentioned in my first review, that I won't repeat here. Units are missing from Fig 5e.

Replies to Reviewers

Reviewer #1:

The authors addressed the questions raised by the reviewers and I suggest to accept it as it is.

Response: We greatly appreciate the reviewer for his/her valuable time and constructive comments in improving the quality of this manuscript.

Reviewer #2:

I read carefully the revised version of the manuscript now entitled « local Cation tuned reversible single-molecule switch in electric double layer. It is much more convincing and its subject is clearly of interest for the readers of Nature communication.

Response: We greatly appreciate the reviewer for his/her positive and constructive comments of our work.

Many of my initial comments have been taken into account. The title has been changed and is now more in line with the findings and the I/V characterization in figure 5f now clearly show a switch between On and Off state. I thus recommend publication in nature communication. However, there is still some important modifications to address prior to publish the paper.

The three following points must be changed.

a) CV of figure 3a show a peak which is attributed to deprotonation/protonation of the carboxyl groups with the help of two references given (ref40 et 41). The author MUST add that there is no faradic processes taking place within this electrochemical potential range. This fact is clearly stated in Reference 41 and even though it is true that in 2008 a similar peak as the one shown here was reported (in reference 40) it remains important to say that this is not linked to any faradic processes. (deprotonation/protonation are not redox events) From reference 41 this is clearly stated the P1 peak is not attributed to protonation/deprotonation but to a reorganization of the ad layers on the surface.

Response: We thank the reviewer very much for the correction, we have revised the “A

linear correlation between current density of oxidation (red square) or reduction (blue square)” to “A linear correlation between anodic (red square) and cathodic (blue square) current peaks”, “total charges of oxidation peaks” to the “total charges of anodic current peaks”. We have also added a description “Due to the good stability of carboxylic acids, no faradaic processes occur in this electrochemical potential range.”

For reference 41, P1 peak at around -0.10 V vs. SCE is attributed to the formation of highly ordered adlayer of TPA (Reorganization of the ad layers) on the surface, which assemble into an infinite 2D structure through hydrogen-bonded structures. In our CV experiments, the current peak occurs at approximately 0 V vs. Pt, which corresponds to 0.48 V vs. SCE, by considering the open circuit potential between the Pt wire and SCE in 0.1 mM 4-MTBA + 50 mM NaClO₄ solution. Thus, the current peak in Figure 3a should be compared to the P2 peak at around 0.5 V vs. SCE, which is also attributed to a potential-induced phase transition including a deprotonation process of the carboxylic acid functional groups, i.e.,*“TPA molecules changed their orientation from flat to vertical or tilted by the deprotonation of one of its carboxylic acid functional groups”* in reference 41. We have added the experimental detail *“The open circuit potential between the Pt wire and SCE is 0.48 V in 0.1 mM 4-MTBA + 50 mM NaClO₄ solution”* in the revised manuscript.

b) I still have problem with the claimed ON/OFF ratio of conductance (or current) Indeed, the paper clearly demonstrate that there is a preferred conductance peak in some electrolyte when the applied potential is positive and that there is no conductance peak (i.e ; no single molecule junction) when the potential is below -0.3 V (see figure 3b for instance) but this does not mean that the ON/OFF ratio exceeds 6.7×10^4 as claimed. Indeed, if we concentrate on the value of the measured current when no SMJs are obtained (at -0.5V), in figure 3b, there is current counts at all current values including for currents higher than that observed for the peak at 0.1 V. So the ratio of current is not 6.7×10^4 and can in fact be anything. If one now looks at the curves given in figure 5f, the current ON is around 4 nA and the current OFF is below 1 nA but is not at all 6.7×10^4 smaller than the current ON. Current Off is not small because there is leakage

and in fact no SMJ are created but the tip is still not far away to the surface and if there is a little bit feedback loop as stated in the caption then the tip will try to move and get closer to the surface and reach the current set point used. In my opinion it is impossible to state that the ON/OFF ratio exceeds four order of magnitude as stated in Figure 5

Response: The point is: when no conductance peak is detected in STM-BJ experiments, can it be said that the conductance value is below the detection limit of the current amplifier? There are two situations in the current study that lead to the disappearance of the conductance peak: One is that the conductance of molecular junction is too small to be detected by current amplifiers in use; Another is that no molecular junction is formed. In our experiments, there is no step feature in the conductance-displacement after rupturing Au atomic contacts in conductance-displacement traces at -0.5 V. With the help of Raman spectroscopy and theoretical simulations, we reveal that the strong molecular carboxyl-metal cation coordination at the negatively charged electrode surface hinders the formation of molecular junctions for electron tunnelling. To avoid this controversy, we have removed the description of the ON/OFF ratio, and added a description “*The conductance peak disappears due to the strong molecular carboxyl-metal cation coordination on the negatively charged electrode surface, which is considered as the conductance OFF state. Thus, the conductance ON/OFF states might be effectively achieved through the electrochemical control.*” in the revised manuscript.

There are current counts at all current values at -0.5 V in figure 3b, including some currents higher than that observed for the peak at 0 V, because the STM-BJ firstly forms metal atomic contacts, then the metal atomic contacts break, and form a nanogap that can trap molecules to form molecular junctions. If molecular junctions are not formed, there will also be an exponentially decaying tunneling current, including some current values higher than the conductance of single-molecule junctions.

We agree that the base current including tip leakage, Faraday and non-Faraday current can affect the current minimum and thus the switching ratio. The I-V traces in Fig. 5f is recorded with STM feedback loop turned off. It is worth pointing out that the I-V traces in Fig. 5f is a dynamic process by simultaneously scanning the potentials of Au tip and substrate. There is an additional charging current of electric double layer to

increase the base current, compared with STM-BJ experiments that performed at a specific potential in quasi-steady state. Thus, the tip current at the OFF state in Fig. 5f is larger.

According to the reviewer's constructive comments, we have removed the statement that ON/OFF ratio exceeds 6.7×10^4 in the revised manuscript, and added a description *"The conductance peak disappears due to the strong molecular carboxyl-metal cation coordination on the negatively charged electrode surface, which is considered as the conductance OFF state. Thus, the conductance ON/OFF states might be effectively achieved through the electrochemical control"* and a discussion *"In addition, it is worth mentioning that this local cation-tuned single-molecule switch depends on the rate at which the applied potential changes the structure of electric double layer. This might lead to low switching frequency. On the other hand, when the gate potential is changed, the base current including the electric double layer charging current and the tip leakage current can affect the switching performance such as ON/OFF ratio"* in the revised manuscript.

c) My last problem is the I(t) curves in Figure 5d. Usually I(t) curves on SMJs are done without any feedback loop on and telegraphic style showing ON/OFF switch with time and (thus able to measure ON/OFF ratio) can be obtained. Such curves must be done without feedback, See for two recent examples (*Nano Letters* 2021, 21 (15), 6540-6548 and *J. Am. Chem. Soc.* 2019, 141, 14788–14797 which could be cited). In Figure 5d a current set point of 4 nA is applied so is impossible to have a current clearly different from 4 nA. Both curves (the red and the green) have been done with a current feedback thus I do not see exactly what they mean. The red curve is centered at 4 nA because of the set point and noisy so I do not exactly see what they mean.

Response: The I(t) test with feedback loop at a fix bias can keep the relative distance between the tip and substrate by a preset current point. As the molecules are trapped into the nanogap between the two electrodes to form molecular junctions, the characteristic electron tunnel-current spikes can be observed on the I-t curves, which have been used to electronic single-molecule identification and sensors (*Nat.*

Nanotechnol., 2010, 5, 868–873; *Nat Commun.*,2016, 7, 13868). This has also been pointed out by Reviewer #3. Thus, we performed I(t) tests with feedback loop at 0 and -0.5 V in Figure 5d (now is Supplementary Fig. 19a) to prove the molecules can be trapped and tethered to two electrodes to form molecular junction at positively charged electrode surface, rather than at negatively charged electrode surface. Fig. 5e (now is Supplementary Fig. 19b) is the statistics of relative height of current spikes in the I-t curves with shifting the baseline of tunnelling current (4 nA) to zero. As Reviewer #3 said, these current spikes in I-t traces could act as a sensor rather than a switch. We have corrected this part in the revised manuscript.

Fig. R1. (a) I-t traces recorded in 0.1 mM 4-MTBA + 50 mM NaClO₄ solution without STM feedback loop at the substrate potentials of 0 and -0.5 V, respectively. Initial tunnelling current setpoint is 0.8 nA under a bias voltage of 50 mV. (b) Zoom in I-t trace showing the formation of single-molecule junction formation.

We have also supplemented the I(t) test without feedback loop according to the reference (*Nano Lett.*, 2021, 21, 6540-6548). The STM tip is stabilized at a fixed distance according to the tunneling parameters ($I_t = 0.8$ nA and $V_{\text{bias}} = 50$ mV). Then we turned off the STM feedback loop and recorded the tip current for 20 s. When no molecule bridges the two electrodes, there is only the base current. Due to thermal, mechanical drift or molecular motions, the tip can contact the molecules, and once a

molecular junction is formed, the current suddenly increases, which means that one or more molecules bridge the two electrodes. Fig. R1 shows the I-t traces as untreated raw data at different substrate potentials. At 0 V, current blinking can be observed at around $t=10$ s, then the current jumps to ON state for lasting more than 1 s, consistent with the previous report (*Nano Lett.*, 2021, 21, 6540-6548). This similar phenomenon repeats around $t=19$ s. The current jump height is about 3.78 nA, corresponding to a conductance of $10^{-3.0} G_0$, which is coherent with single-molecule conductance in the STM-BJ measurements. Instead, only the base current and its fluctuations were observed when the substrate potential was controlled at -0.5 V, indicating no molecular junction formation. These further confirm that the electric field localized cations at different charged electrode surface can modulate the molecule-metal contacts, leading to conductance ON/OFF states.

According to review's comments, we have replaced the Fig. 5d and 5e with Fig. R1 and moved the results and discussions of I-t test with the STM feedback loop to the supplementary information. We have cited the references of *Nano Lett.* 2021, 21, 6540-6548 and *J. Am. Chem. Soc.* 2019, 141, 14788–14797 and added a discussion “we further performed I-t tests without STM feedback loop according to previous reports^{47,48}, The experimental details can be found in Methods section. Fig. 5d shows typical untreated raw I-t traces at different substrate potentials. At 0 V, current blinking can be observed at around $t=10$ s, then the current jumps to ON state for lasting more than 1 s, consistent with the previous report⁴⁷. This similar phenomenon repeats around $t=19$ s. The current jump height is about 3.78 nA, corresponding to a conductance of $10^{-3.0} G_0$, which is coherent with single-molecule conductance in the STM-BJ measurements. Instead, only the base current and its fluctuations were observed when the substrate potential was controlled at -0.5 V. In addition, the I-t tests with a very low STM feedback loop⁴⁹⁻⁵¹ have also shown that the characteristic electron tunnel-current spikes ascribed to trapped molecules in the nanogap of two electrodes to form molecular junction at positively charged electrode surface, rather than at negatively charged electrode surface (Supplementary Fig. 19). These further confirm that the electric field localized cations at different charged electrode surface can modulate the

molecule-metal contacts, leading to conductance ON/OFF states.” in the revised manuscript.

Overall the paper is interesting with a very nice new concept but ON/OFF ratio above 10^4 remains in my opinion not clearly demonstrated in the presented data.

Response: We greatly appreciate the reviewer for his/her valuable time and constructive comments in improving the quality of this manuscript. We have removed the statement that ON/OFF ratio exceeds 6.7×10^4 in the revised manuscript.

Reviewer #3:

The study is improved by enacting the changes. I still find think it is suitable for Nature Communications, as the science is sound and the results are convincing and well-supported by the data, but some issues need to be clarified.

Response: We greatly appreciate the reviewer for his/her positive comments of our work.

Regarding the switch performance I'm a little confused by Fig. 5. Is it correct that the conductance values in Fig. 5a – 5c are taken from fitting to conductance of many junctions followed by averaging? This should be clarified.

Response: Yes, the conductance values in Fig. 5a – 5c are taken from the conductance histograms constructed by thousands of conductance-displacement traces with a Gaussian fit. We have added a description “*The switching cycle tests are summarized in Fig. 5a-c with conductance values taken from the conductance histograms constructed by thousands of conductance-displacement traces with a Gaussian fit*” and “*The conductance peak disappears due to the strong molecular carboxyl-metal cation coordination on the negatively charged electrode surface, which is considered as the conductance OFF state. Thus, the conductance ON/OFF states might be effectively achieved through the electrochemical control.*” in the revised manuscript.

In Fig. 5d the I(t) trace at -0.5 V has a mean value of ~ 4 nA , so if translated into a

Gaussian as done for the trace at 0 V in Fig. 5e it will appear at the same mean value but narrower, as there are no spikes present. I'm not sure what this adds to the switching behaviour, in my mind these I(t) traces at a fixed bias imply the device could act as a sensor rather than a switch, - it reminds me of e.g. <https://www.nature.com/articles/ncomms13868>.

Response: For the statistics of relative height of current spikes in the I-t curves, we have shifted the baseline of tunnelling current (4 nA) to zero. We agree that these current spikes in I-t traces in Fig. 5d (now is Supplementary Fig. 19a) could act as a sensor rather than a switch. We have also supplemented the I(t) test without feedback loop (Fig. R1) according to the previous report (*Nano Lett.*, 2021, 21, 6540-6548). At 0 V, current blinking can be observed at around $t=10$ s, then the current jumps to ON state for lasting more than 1 s, consistent with the previous report (*Nano Lett.*, 2021, 21, 6540-6548). This similar phenomenon repeats around $t=19$ s. The current jump height is about 3.78 nA, corresponding to a conductance of $10^{-3.0} G_0$, which is coherent with single-molecule conductance in the STM-BJ measurements. Instead, only the base current and its fluctuations were observed when the substrate potential was controlled at -0.5 V, indicating no molecular junction formation. These further confirm that the electric field localized cations at different charged electrode surface can modulate the molecule-metal contacts, leading to conductance ON/OFF states.

According to review's suggestion, we have cited the reference of *Nat. Commun.*, 2016, 7, 13868, replaced the Fig. 5d with Fig. R1 and changed the discussion to “we further performed I-t tests without STM feedback loop according to previous reports^{47,48}, The experimental details can be found in Methods section. Fig. 5d shows the typical untreated raw I-t traces at different substrate potentials. At 0 V, current blinking can be observed at around $t=10$ s, then the current jumps to ON state for lasting more than 1 s, consistent with the previous report⁴⁷. This similar phenomenon repeats around $t=19$ s. The current jump height is about 3.78 nA, corresponding to a conductance of $10^{-3.0} G_0$, which is coherent with single-molecule conductance in the STM-BJ measurements. Instead, only the base current and its fluctuations were observed when the substrate potential was controlled at -0.5 V. In addition, the I-t tests with a very low STM

feedback loop⁴⁹⁻⁵¹ have also shown that the characteristic electron tunnel-current spikes ascribed to trapped molecules in the nanogap of two electrodes to form molecular junction at positively charged electrode surface, rather than at negatively charged electrode surface (Supplementary Fig.19). These further confirm that the electric field localized cations at different charged electrode surface can modulate the molecule-metal contacts, leading to conductance ON/OFF states.” in the revised manuscript.

Additionally, regarding the issue of reporting the ON/OFF ratio as 6.7×10^4 – I find removing it from just the abstract a strange course of action. Either it is correct, and can remain wherever, or it is incorrect/dubious, and then should be removed or heavily caveated.

Response: Just like our response to Reviewer 2’s question “b)”, we have removed the description of the ON/OFF ratio of 6.7×10^4 in order to avoid this controversy.

The point is: when no conductance peak is detected in STM-BJ experiments, can it be said that the conductance value is below the detection limit of the current amplifier? There are two situations in the current study that lead to the disappearance of the conductance peak: One is that the conductance of molecular junction is too small to be detected by current amplifiers in use; Another is that no molecular junction is formed. In our experiments, there is no step feature in the conductance-displacement after rupturing Au atomic contacts in conductance-displacement traces at -0.5 V. With the help of Raman spectroscopy and theoretical simulations, we reveal that the strong molecular carboxyl-metal cation coordination at the negatively charged electrode surface hinders the formation of molecular junctions for electron tunnelling. To avoid controversy, we have removed the description of the ON/OFF ratio exceeds 6.7×10^4 in the revised manuscript, and added a description “*The conductance peak disappears due to the strong molecular carboxyl-metal cation coordination on the negatively charged electrode surface, which is considered as the conductance off state. Thus, the conductance ON/OFF states might be effectively achieved through the electrochemical control.*” in the revised manuscript.

Overall, some re-writing or restructuring of the discussion around Fig. 5 should be undertaken to clarify these, discussing limitations, and possibly other potential uses.

Response: We have changed Fig. 5 and rewritten the discussion, and we have also added a discussion to clarify the limits of the local-cation controlled single-molecule switch *“In addition, it is worth mentioning that this local cation-tuned single-molecule switch depends on the rate at which the applied potential changes the structure of electric double layer. This might lead to low switching frequency. On the other hand, when the gate potential is changed, the base current including the electric double layer charging current and the tip leakage current can affect the switching performance such as ON/OFF ratio.”* in the revised manuscript.

Finally, there are still typos remaining, including some that I explicitly mentioned in my first review, that I won't repeat here. Units are missing from Fig 5e.

Response: We thank the reviewer very much for the correction. As noted in the first review, we have corrected 'wieldy' to 'widely', and carefully checked the manuscript and revised typo and format errors marked with yellow.

REVIEWERS' COMMENTS

Reviewer #2 (Remarks to the Author):

The authors have strongly improved the manuscript and all the points raised during the reviewing process have now been well taken into account.

i am glad to recommend publication of the revised manuscript in Nature Communication.

Congratulation to the authors for this nice work.

Reviewer #3 (Remarks to the Author):

I am happy for the paper to be accepted.

Replies to Reviewers

Reviewer #2 (Remarks to the Author):

The authors have strongly improved the manuscript and all the points raised during the reviewing process have now been well taken into account. I am glad to recommend publication of the revised manuscript in Nature Communication.

Congratulation to the authors for this nice work.

Response: We greatly appreciate the reviewer for his/her valuable time and constructive comments in improving the quality of this manuscript.

Reviewer #3 (Remarks to the Author):

I am happy for the paper to be accepted.

Response: We greatly appreciate the reviewer for his/her valuable time and constructive comments in improving the quality of this manuscript.